# CONTAMINATION DETECTION FOR VLMS USING MULTI-MODAL SEMANTIC PERTURBATIONS

**Jaden Park**[1]**, Mu Cai**[1]**, Feng Yao**[2]**, Jingbo Shang**[2]**, Soochahn Lee**[3]**, Yong Jae Lee**[1]

[1]University of Wisconsin-Madison  [2]University of California, San Diego  [3]Kookmin University

## ABSTRACT

Recent advances in Vision–Language Models (VLMs) have achieved state-of-the-art performance on numerous benchmark tasks. However, the use of internet-scale, often proprietary, pretraining corpora raises a critical concern for both practitioners and users: inflated performance due to *test-set leakage*. While prior works have proposed mitigation strategies such as decontamination of pretraining data and benchmark redesign for LLMs, the complementary direction of developing detection methods for *contaminated VLMs* remains underexplored. To address this gap, we deliberately contaminate open-source VLMs on popular benchmarks and show that existing detection approaches either fail outright or exhibit inconsistent behavior. We then propose a novel simple yet effective detection method based on *multi-modal semantic perturbation*, demonstrating that contaminated models fail to generalize under controlled perturbations. Finally, we validate our approach across multiple realistic contamination strategies, confirming its robustness and effectiveness. The code and perturbed dataset are released here: https://github.com/jadenpark0/mm-perturb.

**Multi-modal Semantic Perturbation**

Figure 1: Example of our multi-modal semantic perturbation pipeline applied to RealWorldQA benchmark. Using ControlNet trained with Flux models, a new speed limit sign is generated, changing the correct answer from (B) to (C) while preserving the original image's overall composition. A contaminated model that has memorized the original question is likely to fail on the perturbed version.

# 1 INTRODUCTION

Recent advances in Vision-Language Models (VLMs) have resulted in remarkable performance across a wide range of tasks, including visual reasoning (Yue et al., 2024; Liu et al., 2024b; Chen et al., 2024a), real-world understanding (xAI, 2024), and complex mathematical problems (Zhang et al., 2024b; Lu et al., 2024b). A typical VLM training pipeline involves pretraining a vision encoder and language backbone on internet-scale data, followed by a fine-tuning stage on high-quality multimodal

| Requirements | | Reliability | Practicality | Consistency |
|---|---|:---:|:---:|:---:|
| N-gram Accuracy | (Xu et al., 2024) | ✗ | ✗ | ✗ |
| Shared Likelihood | (Oren et al., 2023) | ✗ | ✗ | ✗ |
| Guided Prompting | (Golchin & Surdeanu, 2024) | ✗ | ✗ | ✗ |
| Multi-modal Leakage | (Chen et al., 2024a) | ✗ | ✗ | ▲ |
| CircularEval | (Liu et al., 2024b) | ▲ | ✗ | ✗ |
| Choice Confusion | (Yao et al., 2024) | ✗ | ✓ | ▲ |
| BGR Shuffling | (Lu et al., 2024a) | ✗ | ✗ | ✗ |
| Image Masking / Option Shuffling | (Song et al., 2025) | ✗ | ✗ | ✗ |
| Multi-modal Semantic Perturbation | **Ours** | ✓ | ✓ | ✓ |

Table 1: Analysis of existing detection methods on VLMs. We label the detection method with ✓ if it satisfies all of Requirement 1, 2 or 3 and with ✗ otherwise. ▲ indicates that the requirement is partially observed but not consistently with varying contamination settings. Most existing detection methods fail to meet the requirements and cannot accurately classify contaminated models. Our method, however, satisfies all requirements. Results for N-gram Accuracy, Shared Likelihood and Guided Prompting are in the Appendix C.

instruction-tuning datasets. However, as these training corpora are often proprietary with their exact composition undisclosed, a critical concern has emerged: public benchmark data may have leaked into the training set, leading to inflated and misleading performance metrics.

Test-set leakage presents a practical and significant challenge. For model users, it becomes difficult to disentangle genuine reasoning and generalization from mere memorization. For developers, exhaustively verifying the absence of test examples in massive pretraining corpora is prohibitively expensive (Bai et al., 2023). Although early works on large models proposed decontamination steps by removing n-gram overlaps (Brown et al., 2020; Abdin et al., 2024a), many recent models do not report such procedures, leaving the extent of contamination largely unexamined.

To address this, several methods have been proposed to detect data contamination. One line of work focuses on *verbatim memorization*, testing whether a model can reconstruct exact benchmark questions with high confidence (Xu et al., 2024; Golchin & Surdeanu, 2024; Oren et al., 2023). Another line of work examines *generalization failures*, measuring if a model that succeeds on an original question fails on variants with similar or easier difficulty, which is interpreted as an evidence of memorization (Mirzadeh et al., 2024; Yao et al., 2024; Huang et al., 2025).

However, these detection methods were primarily designed for Large Language Models (LLMs) and often overlook the unique, multi-modal nature of VLMs. Applying simple text-based perturbations to a VLM may not be sufficient, as the model could rely on visual features that remain unchanged. This discrepancy exposes a critical gap and raises a key question:

> *Is there a reliable, practical, and consistent method for detecting contamination in VLMs?*

In this study, we conduct a systematic analysis of the data contamination problem, from which we derive grounded definitions of the core requirements – reliability, practicality, and consistency. Guided by these definitions, we systematically contaminate open-source VLMs under varying fine-tuning epochs, data composition, and training strategies (e.g., standard fine-tuning vs. LoRA (Hu et al., 2021)). Our results show that existing detection methods struggle with the complexities of VLMs, frequently failing to meet the core requirements across diverse contamination scenarios (see Table 1).

To overcome these limitations, we introduce a novel **multi-modal semantic perturbation** pipeline. Our method generates new test examples by subtly altering the semantics of the *image* while preserving its overall composition, thereby creating variants of comparable or lower difficulty (Figure 1). The core principle is that contaminated models, which have merely memorized an image-text pair, will fail to generalize to this perturbed input despite the similar or lower difficulty of the questions. In contrast, clean models with genuine reasoning capabilities should perform correctly or even better. This approach enables robust contamination detection without requiring any ground-truth knowledge of the leaked data.

Our contributions are threefold:

1. We propose a novel and simple yet effective detection framework based on multi-modal semantic perturbations, which effectively identifies contaminated models by testing for generalization failures in the visual domain.
2. We validate our method across multiple contamination settings, proving that it is reliable, practical and consistent, satisfying all key requirements for a robust detection method.
3. We conduct the first systematic study of VLM behavior under diverse contamination and detection strategies, demonstrating that existing methods designed for LLMs are often unreliable for detecting contaminated VLMs.

## 2 INVESTIGATION SETUP

This section establishes the formal framework for our analysis of data contamination. We begin by defining the degree of contamination, state our core assumption about its relationship with model generalization, and finally, outline three essential requirements for a robust detection method.

Our analysis is built upon a formal definition of contamination at the data-point level. For a given data point $x$ of a dataset $\mathcal{D}$ and a training process consisting of $n$ epochs, we define:

> **Definition 1.** (Degree of Contamination). The *degree of contamination* for a data point $x$ is:
>
> $$\deg_{\mathcal{D}}(x) = \left( \sum_{d \in \mathcal{D}} \mathbf{1}_{\{x=d\}} \right) \times n.$$
>
> This quantity reflects the total number of times $x$ is seen during training. In our experimental setup, where a model $\mathcal{M}$ is fine-tuned on the entire benchmark dataset $\mathcal{D}$ for $n$ epochs, this simplifies to $\deg_{\mathcal{D}}(\mathcal{M}) = n$.

This definition motivates our central assumption, grounded in prior work on model memorization (Zhang et al., 2017; Carlini et al., 2021; Kandpal et al., 2023):

> **Assumption 1.** Data points with a higher degree of contamination are more likely to be memorized, which increases overfitting risk and impairs generalization.

In particular, we posit that as the degree of contamination grows, models will exhibit degraded performance on perturbed or out-of-distribution variants of benchmark items, even when the original questions are answered correctly. The effect need not scale linearly, since fine-tuning can disproportionately distort embedding spaces (Choi et al., 2025).

Based on Assumption 1, we argue that any practical and effective detection method must satisfy three fundamental requirements:

> **Requirement 1.** (Practicality). The method must operate without assuming any prior knowledge of clean model (e.g. the model's training corpus), relying only on black-box interactions.
>
> **Requirement 2.** (Reliability). The method must detect contaminated models across heterogeneous fine-tuning strategies (e.g., standard fine-tuning vs. LoRA).
>
> **Requirement 3.** (Consistency). The method's detection signal should be positively correlated with degree of contamination $n = \deg_{\mathcal{D}}(\mathcal{M})$.

Together, these requirements define a principled framework for evaluating contamination detection. A method that satisfies all of them can practically flag contaminated models without the knowledge of clean model's behavior, remain agnostic to training specifics, and provide a signal proportional to the extent of contamination.

## 3 PREPARATION OF CONTAMINATED MODELS

**Models and benchmarks.** To analyze contamination detection across diverse settings, we pair complementary model families and benchmarks. On the model side, we use LLaVA-v1.5-7B (Liu et al., 2024a) – a LLaMA–adapter design – and Qwen2-VL-7B (Wang et al., 2024), which integrates tighter multimodal alignment. Both are widely adopted open-source VLMs with publicly documented training details (e.g., released corpora or decontamination policies, though we do not rely on this for detection), making them suitable for controlled contamination experiments.

For benchmarks, we use MMStar (Chen et al., 2024a) and RealWorldQA (xAI, 2024), which consist of questions that strictly require visual *and* textual evidence. MMStar aggregates and filters prior multi-modal benchmarks by removing questions that are leaked or do not have visual dependence, while RealWorldQA enforces visual dependence by design. This prevents linguistic shortcuts and ensures our perturbation-based detector evaluates genuine multi-modal reasoning.

**Training strategies.** We contaminate models by training directly on evaluation data. Because VLM training typically has two stages—(i) large-scale pretraining of the vision encoder and language backbone and (ii) instruction-tuned multimodal fine-tuning—leakage can arise at either stage. Ablating pretraining is computationally prohibitive, so we focus on *continual fine-tuning*, which affords precise control over contamination levels.

We compare standard fine-tuning and LoRA (Hu et al., 2021). For standard fine-tuning, we follow three common variants: fine-tune the LLM and adapter as in LLaVA-v1.5-7B; fine-tune only the LLM as in Qwen2-VL-7B; or unfreeze all parameters as in InternVL (Chen et al., 2024b). This diversity tests robustness across parameter-efficient and full fine-tuning regimes.

**Epochs.** We save checkpoints at epochs 1–3 (i.e. $\deg_{\mathcal{D}}(\mathcal{M}) \in \{1, 2, 3\}$), since VLMs are typically fine-tuned for at most three epochs – and often just one. Contamination footprints of varying strength enables a graded analysis of detection sensitivity (Liu et al., 2024a; Chen et al., 2025).

**Hyperparameters.** We use model defaults, adjusting only the learning rate to ensure inflated performance. For LLaVA-v1.5-7B, we follow the official repo; for Qwen2-VL-7B, we use LLaMA-Factory (Zheng et al., 2024) to train the models. (Full settings are in the Appendix A.1.)

## 4 MULTI-MODAL SEMANTIC PERTURBATION

We propose multi-modal semantic perturbation framework for contamination detection, that generates variants of image–text pairs with similar or lower difficulty while keeping the original image composition intact. Our method consistently detects contaminated models and satisfies all three requirements, while existing methods fail to yield a stable signal of test-set leakage in VLMs.

To generate semantically perturbed questions, we combine an LLM with a diffusion-based generative model. In our main experiments, we use GPT-4o (OpenAI, 2024) and Flux (Labs, 2024) + ControlNet (Zhang et al., 2023), but later show that our framework is model-agnostic (Section 7).

The pipeline is as follows. First, we randomly change the answer of the original question to a different option, preventing contaminated models from getting away with memorized responses. Next, GPT-4o generates a dense caption of the image, conditioned both on the original question and the newly chosen answer choice. We find this explicit conditioning essential: it ensures the caption highlights the salient visual features that drive the perturbation process, allowing the generation of questions of similar or lower difficulty (Section 6). Flux ControlNet then uses this caption, together with Canny edge maps (Canny, 1986), to guide the diffusion process. ControlNet preserves the global structure of the image while introducing new elements that *minimally* alter semantics, yielding an updated image with the randomly sampled different correct answer (Figure 2). We note that, due to limitations in rendering text or complex geometries, especially at low resolution, some generated images do not fully correspond to the new answer. To mitigate this, we filter generated pairs using a single criterion: perturbed questions must be answerable unambiguously.[1] For the main results, we apply manual

---

[1]Generation quality itself is not considered. This ensures that the evaluation set focuses solely on reasoning rather than visual fidelity. While our main results report human-filtered outcomes, we show in Table 9 that this step can be automated. We stress that filtering is necessitated by current generative model limitations, not by our detection principle.

filtering to demonstrate the upper-bound performance of our approach. However, as shown in Table 9, this manual step can be replaced with automated filtering with a strong reasoning model.

To detect whether a model is contaminated on a dataset-level, i.e. has memorized the training image-text pairs, we compare its aggregate performance (i.e. accuracy) on the original vs. perturbed benchmarks. If the model fails to generalize (i.e. it gets the original input correct but the perturbed one incorrect, leading to a lower performance on the perturbed benchmark), we flag contamination[2]. This is because a clean model with genuine reasoning capabilities should perform correctly in both settings, given that they have comparable difficulty. Critically, our approach enables robust contamination detection without requiring any ground-truth knowledge of the clean model's behavior.

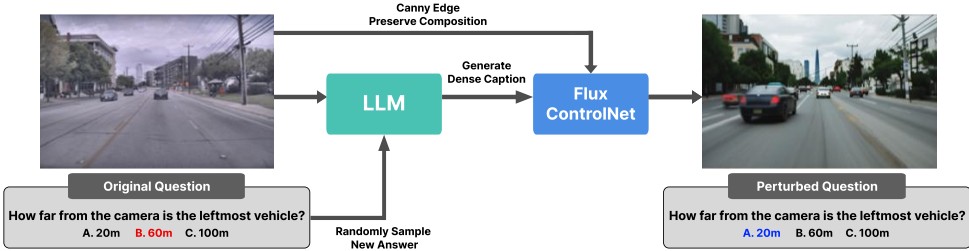

Figure 2: Illustration of our multi-modal semantic perturbation pipeline. The original question–image pair is used to generate a dense caption with an LLM, which guides Flux ControlNet to produce a perturbed image and new answer, yielding a modified but semantically consistent benchmark sample.

## 5   COMPARATIVE EVALUATION OF CONTAMINATION DETECTION

In this section, we present our main results. After perturbation and filtering, 440 image–question pairs remain from the original 765 in RealWorldQA, and 478 remain from 1,500 in MMStar (Section 6 shows that these subsets remain representative of the full datasets). Table 2 reports results for MMStar; results for RealWorldQA are in in the Appendix (Table 13).

First, we see that clean models perform better than contaminated models, confirming that the perturbed questions are indeed of equal or lower difficulty, and that *we are testing against perturbations that the clean models are robust against*. In contrast, *all* contaminated models perform worse, enabling our method to reliably detect via a simple check for performance drops.

We compare to the following contamination detection methods:

**Multi-modal leakage** (Chen et al., 2024a). With multi-modal leakage, contamination is flagged by measuring text-only performance on benchmarks that require visual input. Any boost in text-only performance after fine-tuning indicates memorization of leaked problems rather than genuine multi-modal reasoning. Multi-modal leakage requires clean models by design and hence does not satisfy Practicality (Req. 1). From Table 2 we observe that Reliability (Req. 2) is not satisfied since it fails to detect LLaVA-v1.5-7B trained for 3 epochs with standard fine-tuning. While the performance gain positively correlates with the degree of contamination in other cases, this breaks across benchmarks and training strategies such as for Qwen2-VL-7B trained for 3 epochs with LoRA and LLaVA-v1.5-7B trained for 3 epochs with standard fine-tuning, only partially satisfying Consistency (Req. 3).

**CircularEval** (Liu et al., 2024b). Selection bias in multiple-choice questions is mitigated by rotating answer options $n$ times and requiring the model to be correct on all rotations. This stricter criterion lowers absolute accuracy compared to standard evaluation. For contamination detection, however, CircularEval's effectiveness is limited. Practicality (Req. 1) fails, as CircularEval lacks a clear threshold-independent detection mechanism. Reliability (Req. 2) is undermined by inconsistent contamination signals, as it fails to detect LLaVA-v1.5-7B trained with LoRA for 2 and 3 epochs, and Qwen2-VL-7B trained with standard fine-tuning for 2 epochs. And despite some positive trends, Consistency (Req. 3) breaks across models and training setups, as shown with LLaVA-v1.5-7B trained with LoRA for 2, 3 epochs and Qwen2-VL-7B trained with standard fine-tuning for 2,3 epochs, limiting its diagnostic value.

---

[2]We make a note that our approach can also be used to detect contamination on a **sample-level** by comparing the model's behavior on the perturbed sample instead of the aggregate performance.

| Method | Metric | LLaVA-v1.5-7B (clean) | LoRA (contaminated) | | | LLM+MLP (contaminated) | | | Require clean model? |
|---|---|---|---|---|---|---|---|---|---|
| | | — | Epoch 1 | Epoch 2 | Epoch 3 | Epoch 1 | Epoch 2 | Epoch 3 | (Practicality (Req. 1)) |
| **Ours** | **MMStar** | 37.78 | 52.53 | 50.71 | 54.34 | 41.82 | 48.89 | 50.71 | No ✓ |
| | **MMStar_P** | 69.29 | 44.24 | 37.58 | 38.18 | 33.33 | 37.37 | 36.97 | |
| | Δ | +31.51 | –8.29 | –13.13 | –16.16 | –8.49 | –11.52 | –13.74 | |
| | **Success?** | ✓ | ✓ | ✓ | ✓ | ✓ | ✓ | ✓ | |
| **CircularEval** | **MMStar** | 37.78 | 52.53 | 50.71 | 54.34 | 41.82 | 48.89 | 50.71 | Yes ✗ |
| | **MMStar_C** | 26.06 | 29.09 | 45.66 | 55.56 | 25.86 | 22.83 | 22.02 | |
| | Δ | -11.72 | -23.44 | -5.05 | +1.22 | -15.96 | -26.06 | -28.69 | |
| | **Success?** | – | ✓ | ✗ | ✗ | ✓ | ✓ | ✓ | |
| **Choice Confusion** | **MMStar** | 37.78 | 52.53 | 50.71 | 54.34 | 41.82 | 48.89 | 50.71 | No ✓ |
| | **MMStar_G** | 71.92 | 53.54 | 65.66 | 69.09 | 62.83 | 66.26 | 62.83 | |
| | Δ | +34.14 | +1.01 | +14.95 | +14.75 | +21.01 | +17.37 | +12.12 | |
| | **Success?** | ✓ | ✗ | ✗ | ✗ | ✗ | ✗ | ✗ | |
| **Multi-modal Leakage** | **MMStar_to** | 19.39 | 26.67 | 29.70 | 30.51 | 19.80 | 26.87 | 8.69 | Yes ✗ |
| | Δ | – | +7.28 | +10.31 | +11.12 | +0.41 | +7.48 | -10.70 | |
| | **Success?** | – | ✓ | ✓ | ✓ | ✓ | ✓ | ✗ | |

| Method | Metric | Qwen2-VL-7B (clean) | LoRA (contaminated) | | | LLM only (contaminated) | | | Require clean model? |
|---|---|---|---|---|---|---|---|---|---|
| | | — | Epoch 1 | Epoch 2 | Epoch 3 | Epoch 1 | Epoch 2 | Epoch 3 | (Practicality (Req. 1)) |
| **Ours** | **MMStar** | 62.02 | 78.38 | 94.14 | 95.96 | 89.90 | 97.98 | 98.99 | No ✓ |
| | **MMStar_P** | 78.18 | 71.31 | 65.25 | 63.64 | 60.40 | 54.95 | 55.96 | |
| | Δ | +16.16 | –7.07 | –28.89 | –32.32 | –29.50 | –43.03 | –43.03 | |
| | **Success?** | ✓ | ✓ | ✓ | ✓ | ✓ | ✓ | ✓ | |
| **CircularEval** | **MMStar** | 62.02 | 78.38 | 94.14 | 95.96 | 89.90 | 97.98 | 98.99 | Yes ✗ |
| | **MMStar_C** | 55.96 | 54.95 | 55.56 | 55.76 | 82.63 | 91.92 | 92.73 | |
| | Δ | -6.06 | -23.43 | -38.58 | -40.20 | -7.27 | -6.06 | -6.26 | |
| | **Success?** | – | ✓ | ✓ | ✓ | ✓ | ✗ | ✓ | |
| **Choice Confusion** | **MMStar** | 62.02 | 78.38 | 94.14 | 95.96 | 89.90 | 97.98 | 98.99 | No ✓ |
| | **MMStar_G** | 94.34 | 94.34 | 93.13 | 93.13 | 96.77 | 96.77 | 97.78 | |
| | Δ | +32.32 | +15.96 | -1.01 | -2.83 | +6.87 | -1.21 | -1.21 | |
| | **Success?** | ✓ | ✗ | ✓ | ✓ | ✗ | ✓ | ✓ | |
| **Multi-modal Leakage** | **MMStar_to** | 22.83 | 27.07 | 28.48 | 28.08 | 44.24 | 51.31 | 52.73 | Yes ✗ |
| | Δ | – | +4.24 | +5.65 | +5.25 | +21.41 | +28.48 | +29.90 | |
| | **Success?** | – | ✓ | ✓ | ✓ | ✓ | ✓ | ✓ | |

Table 2: Performance of LLaVA-v1.5-7B (top) and Qwen2-VL-7B (bottom) on the MMStar dataset (Corresponding RealWorldQA results are in the Appendix 13). We compare to "Multi-modal Leakage" Chen et al. (2024a), CircularEval (Liu et al., 2024b), and "Choice Confusion" Yao et al. (2024). Clean models perform better on our perturbed dataset – confirming that the perturbed questions are indeed of equal or lower difficulty. In contrast, all contaminated models perform worse, enabling reliable detection by our method via a simple check for performance drops. "_P" denotes the semantically perturbed version; "to" denotes text-only performance; "_C" denotes evaluation using circular options; "_G" denotes evaluation using choice confusion; Δ denotes the difference in performance, with positive values indicating gains. "Success?" indicates whether the method detected contamination. "Require clean model?" indicates whether the method requires access to a clean model as a baseline. If a clean model is required as reference, the method cannot be used to detect the reference models themselves, so the entry is marked as "–". Full results of multi-modal semantic perturbation is listed in Appendix F.

**Choice confusion** (Yao et al., 2024). Generalization is tested by constructing an easier benchmark variant: false options are replaced with correct answers drawn from unrelated questions. Clean models should leverage this simplification to improve, whereas contaminated models – bound by memorized original answers – might not. We apply this method to measure the model's performance drop (Δ) between the original and generalized versions. Clean models gain substantially on generalized benchmarks – up to +34.04 on MMStar and +21.30 on RQA – confirming their generalization ability. By contrast, contaminated variants show much smaller gains or even losses. This failure to benefit from semantically irrelevant but easier choices reflects classic memorization-based contamination. As a detection method, choice confusion meets Practicality (Req. 1) since the method detects models that perform worse on the generalized benchmark, but its sensitivity varies across fine-tuning regimes. Choice confusion fails to detect LLaVA-v1.5-7B regardless of its training strategy or the number of epochs, and for LLaVA-v1.5-7B trained with LoRA, the model shows improved performance as the number of training epochs increases. Hence the method fails to satisfy Reliability (Req. 2) while Consistency (Req. 3) is partially observed in other training strategies.

Importantly, unlike other approaches, our method requires no dataset-specific thresholds or prior knowledge of leaked data, satisfying Practicality (Req. 1). Moreover, the performance drop scales with contamination degree, satisfying Consistency (Req. 3), and persists across all training regimes – standard fine-tuning, LoRA, and full parameter unfreezing, satisfying Reliability (Req. 2).

## 6    ANALYSIS OF MULTI-MODAL SEMANTIC PERTURBATION

**Filtered images form a representative subsample.** Since manual filtering removes a substantial portion of the original data, a natural concern is whether the filtered sets remain representative. To test this, we first compare model performance on the full and filtered datasets and find that the results closely align (Table 3). This confirms that the filtering step does not introduce systematic bias, and the remaining subsets preserve the distributional properties of the original benchmarks.

| Model | RQA (765 imgs) | RQA_filtered (440 imgs) | MMStar (1500 imgs) | MMStar_filtered (495 imgs) |
|---|---|---|---|---|
| **LLaVA-v1.5-7B** | 49.01% | 52.05% | 32.87% | 37.78% |
| **Qwen2-VL-7B** | 70.33% | 70.45% | 59.80% | 61.62% |

Table 3: Performance of clean models on RealWorldQA (RQA) and MMStar before and after filtering.

**Why perturbations yield a generalized benchmark.** Our core assumption is that by preserving the original question and only altering the answer choice, the question difficulty remains comparable, and that clean models that answer the original question correctly will solve the variant. In addition, we often observe that the perturbation highlights salient visual cues more clearly than the original images (Fig. 3), collectively yielding an alternate benchmark that is similar or easier. We validate this empirically: clean models consistently achieve higher accuracy on perturbed benchmarks (Table 2).

| **Question** | What does the text on the traffic sign say? | A. Student | B. Children | C. Police |
|---|---|---|---|---|

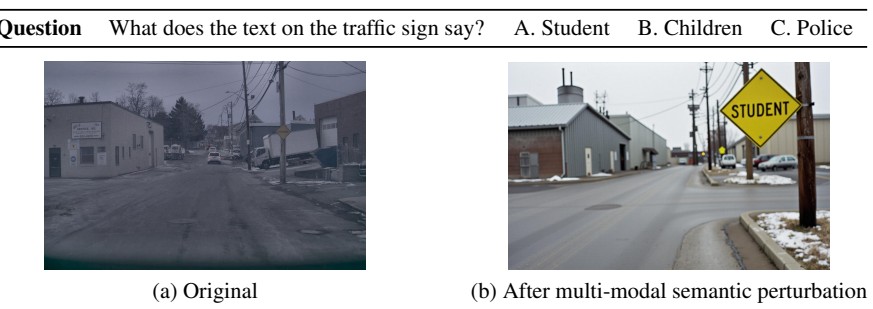

(a) Original          (b) After multi-modal semantic perturbation

Figure 3: Example where the perturbed variant is easier to solve than the original. In the original image, the traffic sign is small and the text barely legible; after perturbation, the sign is enlarged and clearly visible.

**Providing the question-answer pair in caption generation is critical.** As described in Sec. 4, conditioning the caption generation on both the original question, original answer and the new answer is critical to creating a generalized benchmark. One natural approach is to create a variant of the original image from simply conditioning the generated caption on the question and the new correct answer. When we evaluated clean models on images generated from this version of the prompt, the clean model performance was much lower compared to both the original dataset and our final perturbed dataset, with invalid perturbations appearing much more frequently, for example, due to critical component of the image being left out so the question is ambiguous or no longer solvable. Intuitively, the captioning model first reasons about which parts of the image need to change to make the new answer correct, and produces a detailed caption emphasizing those changes. Flux+ControlNet then focuses on rendering these components exactly. Finally, changing the answer is necessary to delineate contaminated models' behavior from simply memorizing the answer.

**Failure modes of multi-modal semantic perturbation.** There are cases where multi-modal perturbation fails to reveal contamination. The perturbed image may differ in its visual details that it no longer closely resembles the original. In such cases, contaminated models may answer both the original and perturbed questions correctly as shown in Figure 4, hiding the contamination of the model. To ensure that such failure cases are rare, we manually inspected the perturbed datasets and verified that only 8 out of 440 images ($\sim$1.8%) from perturbed RealWorldQA and 17 out of 495 images ($\sim$3.4%) from perturbed MMStar deviate from the original question's visual details.

| **Question** | Which vehicle is closer to us, the school bus or the black SUV? |
| | A. School bus  B. Black SUV  C. They are at the same distance. |

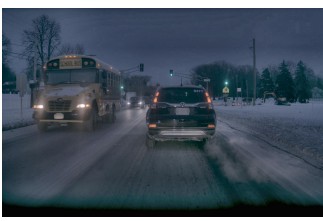
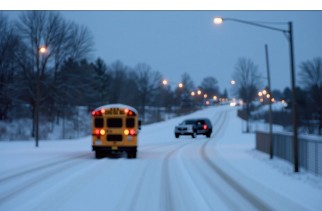

(a) Original                    (b) After multi-modal semantic perturbation

Figure 4: Example where a contaminated model answers both the original and perturbed questions correctly. This may occur when visual details change significantly that the perturbed image no longer closely resembles the original.

**Limitations of multi-modal semantic perturbation framework.** For semantic perturbation to be valid, questions must enforce visual dependence. If a question can be answered without visual input, perturbing the image is meaningless, and altering only the answer invalidates the task. As such, we restricted our study to RealWorldQA and MMStar, which are VQA benchmarks with strict visual grounding. This constraint highlights an important boundary: our method is only effective when visual semantics directly determines the correct answer.

Both RealWorldQA and MMStar are multiple-choice benchmarks. Although the multiple-choice format simplifies evaluation, our framework is not inherently tied to this setting. Once visual evidence is perturbed, evaluation can be adapted to free-form tasks using string matching, likelihood-based scoring, or LLM-as-a-judge approaches. This principle extends naturally beyond multi-choice VQA, such as free-form QA settings, as shown in Appendix F.7.

Finally, manual filtering is required only because of current limitations of diffusion models. As generative models improve, we expect manual filtering to become unnecessary.

## 7  ABLATION STUDIES

We next validate that the core idea – testing for memorization via *mutli-modal semantic perturbations* – is robust to design choices. Specifically, we show our method is not tied to synthetic edits, generalizes to paraphrased data, scales to larger models and alternate contamination regimes, and works without GPT-4o or manual filtering.

**Real-world counterfactuals.** NaturalBench (Li et al., 2024) provides real counterfactual pairs – photos of the same scene under altered conditions – serving as a natural analogue to our synthetic variants. We fine-tune on one variant and evaluate on its paired counterfactual. Contaminated models drop sharply (up to 45.95%), while clean models remain comparatively stable (Table 6), showing our method generalizes beyond synthetic perturbations[3]. Thus, any reliable semantic variation – natural, procedural, or synthetic – fits our framework. Full results are listed in F.2.

**Contamination with paraphrased data.** A simple but effective contamination is paraphrasing the data the model is contaminated on. To test whether models contaminated with paraphrased data can be detected, we paraphrase the questions using GPT-4o with minimal n-gram overlap. The contaminated models are then evaluated on the original benchmark and its perturbed variant. Table 4 and 5 show that contaminated models show inflated performance on original benchmarks with lowered performance on perturbed benchmarks, proving our detection method is robust against paraphrasing.

**Model scale: LLaVA-13B.** To assess scalability, we repeat the RealWorldQA experiment with LLaVA-v1.5-13B. Table 7 shows our detector remains effective at this scale. Given that larger models are more prone to memorization, these findings indicate the approach is suitable for stronger VLMs. Full results are listed in F.2.

**Test-set leakage during pretraining.** We simulate pretraining leakage by mixing RealWorldQA into the 665K instruction-following pretraining corpus and training LLaVA-v1.5-7B for one epoch (Table 8). Using the same 440 filtered images, our method flags contamination, demonstrating applicability beyond fine-tuning-only settings.

---

[3]Note that because counterfactuals are not guaranteed to be easier, clean-model deltas need not be positive.

| Method | Epoch | RQA | RQA_P | Δ |
|---|---|---|---|---|
| **Contamination with paraphrased RealWorldQA** | | | | |
| **LLaVA-v1.5-7B (clean)** | — | 52.05 | 56.36 | +4.31 |
| **LoRA (contaminated)** | 1 | 52.03 | 40.00 | -12.03 |
| | 2 | 55.91 | 40.00 | -15.91 |
| | 3 | 59.09 | 38.86 | -20.23 |
| **LLM+MLP (contaminated)** | 1 | 56.14 | 53.41 | -2.73 |
| | 2 | 61.36 | 48.64 | -12.72 |
| | 3 | 63.64 | 49.77 | -13.87 |

Table 4: Performance of LLaVA-v1.5-7B and contaminated variants on RealWorldQA when trained on the paraphrased benchmark. "_P" denotes the semantically perturbed version. For full results, refer to Appendix F.5.

| Method | Epoch | MMStar | MMStar_P | Δ |
|---|---|---|---|---|
| **Contamination with paraphrased MMStar** | | | | |
| **LLaVA-v1.5-7B (clean)** | — | 37.78 | 68.29 | +31.51 |
| **LoRA (contaminated)** | 1 | 51.52 | 43.64 | -7.88 |
| | 2 | 59.60 | 41.82 | -17.78 |
| | 3 | 62.22 | 40.81 | -21.41 |
| **LLM+MLP (contaminated)** | 1 | 47.27 | 46.06 | -1.13 |
| | 2 | 57.60 | 53.54 | -3.06 |
| | 3 | 62.42 | 57.98 | -4.44 |

Table 5: Performance of LLaVA-v1.5-7B and contaminated variants on MMStar when trained on the paraphrased benchmark. "_P" denotes the semantically perturbed version. For full results, refer to Appendix F.5.

| Method | Epoch | Train Set (%) | Test Set (%) | Δ |
|---|---|---|---|---|
| **Contamination with NaturalBench** | | | | |
| **LLaVA-v1.5-7B (clean)** | — | 65.63 | 65.89 | +0.26 |
| **LoRA (contaminated)** | 1 | 81.53 | 61.37 | −20.16 |
| | 2 | 89.79 | 57.16 | −32.63 |
| | 3 | 91.11 | 57.32 | −33.79 |
| **LLM+MLP (contaminated)** | 1 | 79.95 | 58.79 | −21.16 |
| | 2 | 97.05 | 54.32 | −42.74 |
| | 3 | 98.63 | 53.05 | −45.58 |

Table 6: Performance of clean and contaminated models on NaturalBench. While clean model shows comparable performance on the test set, contaminated models fail to generalize, with performance drop upto -45.58%. For full results, refer to Appendix F.2.

| Method | Epoch | RQA | RQA_P | Δ |
|---|---|---|---|---|
| **Contamination with RealWorldQA** | | | | |
| **LLaVA-v1.5-13B (clean)** | — | 51.14 | 57.27 | +6.13 |
| **LoRA (contaminated)** | 1 | 74.32 | 38.18 | −36.14 |
| | 2 | 73.18 | 32.73 | −40.45 |
| | 3 | 77.05 | 34.77 | −42.28 |
| **LLM+MLP (contaminated)** | 1 | 56.59 | 37.50 | −19.09 |
| | 2 | 71.59 | 38.86 | −32.73 |
| | 3 | 75.45 | 37.27 | −38.18 |

Table 7: Performance of clean and contaminated LLaVA-v1.5-13B on RealWorldQA after multi-modal semantic perturbation. The result satisfies all three requirements. For full results, refer to Appendix F.2.

**Test-set leakage with a mixture of benchmarks** We test our perturbation pipeline under a more realistic contamination scenario: a mixture of other benchmarks. We fine-tune the models with popular multi-modal benchmarks: MathVista Lu et al. (2024b), MMMU Yue et al. (2024), MMBench Liu et al. (2024b) and CV-Bench Tong et al. (2024), resulting in 11,280 image-question pairs, with RealWorldQA and MMStar now consisting only of ∼6.7% and ∼13.3%, respectively. We still observe a clean and consistent detection across all contaminated models, demonstrating that our perturbation pipeline can reliably detect contaminated models even when the contamination signal is weaker. Results are in Appendix F.6 due to limited space.

Finally, we ablate two key components of our pipeline to show modularity of our pipeline.

**Automation of filtering process.** We replace manual validation with the o3 model to filter generated image–question pairs. The automatic pass removes 471 items and retains 294, of which 253 overlap with the manually kept set, indicating high agreement (Table 9). The exact prompt is provided in Appendix B and full results are listed in F.3.

**Caption generation with Molmo-7B-D.** To decouple the pipeline from GPT-4o, we substitute captioning with the lightweight open-source Molmo-7B-D (Deitke et al., 2024). After manual filtering, this variant yields 398 valid pairs and preserves the same detection trends, underscoring the flexibility of our approach (Table 10). Full results are listed in F.4.

# 8 RELATED WORK

**Data contamination in LLMs.** Brown et al. (2020) was among the first to highlight the problem of data contamination when training models on internet-scale corpora, proposing an n-gram overlap–based decontamination technique that was later adopted in Bai et al. (2023). Building on this, several detection methods exploit verbatim memorization to identify leaked test-set examples (Oren et al., 2023; Golchin & Surdeanu, 2024; Xu et al., 2024). More specifically, Xu et al. (2024); Golchin & Surdeanu (2024) test whether the model can accurately reconstruct masked spans of test questions, while Oren et al. (2023) measure the log-probability of the original ordering of multiple-choice options. Choi et al. (2025) instead examine embedding divergence after fine-tuning, exploiting the observation that embeddings of unseen samples change more substantially than those of memorized

| Model | RQA | RQA_P | $\Delta$ |
|---|---|---|---|
| LLaVA-v1.5-7B (clean) | 52.05 | 56.36 | +4.31 |
| Pretrain (contaminated) | 51.82 | 50.00 | −1.82 |

Table 8: Performance of clean vs. pretrained-contaminated models on RealWorldQA (440 images).

| Method | Epoch | RQA | RQA_P | $\Delta$ |
|---|---|---|---|---|
| **Contamination with RealWorldQA** | | | | |
| **LLaVA-v1.5-7B (clean)** | — | 50.68 | 59.86 | +9.18 |
| **LoRA (contaminated)** | 1 | 50.34 | 44.90 | −5.44 |
| | 2 | 66.33 | 46.60 | −19.73 |
| | 3 | 77.21 | 47.28 | −29.93 |
| **LLM+MLP (contaminated)** | 1 | 60.54 | 54.42 | −6.12 |
| | 2 | 63.95 | 55.10 | −8.85 |
| | 3 | 64.63 | 53.40 | −11.23 |

| Method | Epoch | RQA | RQA_P | $\Delta$ |
|---|---|---|---|---|
| **Contamination with RealWorldQA** | | | | |
| **LLaVA-v1.5-7B (clean)** | — | 44.22 | 52.01 | +7.79 |
| **LoRA (contaminated)** | 1 | 44.97 | 35.93 | −9.04 |
| | 2 | 55.78 | 32.66 | −23.12 |
| | 3 | 70.35 | 32.91 | −37.44 |
| **LLM+MLP (contaminated)** | 1 | 56.53 | 39.95 | −16.58 |
| | 2 | 60.80 | 39.45 | −21.35 |
| | 3 | 61.56 | 39.95 | −21.61 |

Table 9: Performance after filtering with the o3 model, resulting in 294 valid image–question pairs. Results satisfy all requirements. For full results, refer to Appendix F.3.

Table 10: Performance after generating and filtering images with captions from Molmo-7B-D, resulting in 398 valid image–question pairs. Results satisfy all requirements. For full results, refer to Appendix F.4.

ones. However, these methods face key limitations: Oren et al. (2023); Golchin & Surdeanu (2024); Xu et al. (2024) perform poorly on contaminated VLMs, while Choi et al. (2025) requires ground-truth clean model behavior to establish a detection threshold, reducing its practicality.

**Generalized benchmarks for detecting contamination.** Yao et al. (2024) tests for generalization rather than memorization. They construct trivial variants of benchmark questions by replacing incorrect multiple-choice options with irrelevant answers. Contaminated models often fail on the easier variants, exposing strong memorization. However, their setup involves training for 36 epochs on a single benchmark – an unrealistic scenario for modern large-scale training. Similarly, GSM-Symbolic and MATH-Perturb (Mirzadeh et al., 2024; Huang et al., 2025) introduce perturbation-based approaches, measuring performance drops on variant questions as contamination signals.

**Contamination detection in VLMs.** VLMs differ from LLMs due to their multi-modal inputs and multi-stage alignment training, which yield distinct contamination dynamics. Lu et al. (2024a) proposed shuffling BGR color channels to mitigate spurious cues, while Song et al. (2025) introduced image masking and option shuffling to test robustness. Yet these approaches struggle to reliably detect contaminated VLMs, as observed performance drops may arise from confounding factors such as visual artifacts or biased sampling rather than true memorization. In contrast, our framework provides consistent detection across diverse fine-tuning regimes, requires no access to leaked data, and yields performance drops that correlate strongly with the degree of contamination.

# 9  CONCLUSION

Recent advances in Vision-Language Models (VLMs) have raised concerns about inflated benchmark performance due to test-set leakage from large-scale, proprietary training corpora. To overcome the limitations of existing detection methods, we introduce a novel approach based on *multi-modal semantic perturbation*. By deliberately contaminating open-source VLMs and evaluating their generalization behavior, we show that our method consistently identifies contamination where prior approaches either fail or yield inconsistent signals. These results establish multi-modal semantic perturbation as a simple and reliable framework for detecting test-set leakage in VLMs.

**Broader impacts.** Our multi-modal semantic perturbation method aims to uncover and quantify the degree of data contamination in VLMs, promoting cleaner training pipelines and more trustworthy models. By systematically characterizing contamination behaviors, it lays the groundwork for robust model evaluation and contamination detection. While these insights could inform adversaries seeking subtler contamination schemes, we believe this work will foster the development of stronger defenses and decontamination strategies.

**Reproducibility statement.** We publicly release our code, models and data.

**LLM usage.** Gemini and ChatGPT were used to polish some of the paper's writing.

**Acknowledgement.** This work was supported in part by NSF IIS2404180, Microsoft Accelerate Foundation Models Research Program, Institute of Information & communications Technology Planning & Evaluation (IITP) grants funded by the Korea government (MSIT) (No. 2022-0-00871, Development of AI Autonomy and Knowledge Enhancement for AI Agent Collaboration), (No. RS-2022-00187238, Development of Large Korean Language Model Technology for Efficient Pre-training), and by the Institute of Information & Communications Technology Planning & Evaluation(IITP) grant funded by the Korea government(MSIT) (RS-2025-02219317, AI Star Fellowship(Kookmin University). The authors would like to thank SeungEun Chung, YoHan Ban, Samuel Low Yu Hang and Junxia Cui for assistance with experiments and Anirudh Sundara Rajan and Zhuoran Yu for helpful discussion.

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

APPENDIX

## A EXPERIMENT SETTINGS

In this section, we detail all hyperparameter settings for contaminating models and for multi-modal semantic perturbation of benchmarks.

### A.1 MODEL TRAINING

We contaminate LLaVA-v1.5-7B by following the official repository [4] and Qwen2-VL-7B using LLaMA-Factory. All settings remain identical except for the learning rates, which were tuned over the ranges shown in Table 11.

| Hyperparameter | LLaVA-v1.5-7B | | Qwen2-VL-7B-Instruct | |
|---|---|---|---|---|
| | Standard | LoRA | Standard | LoRA |
| Learning rate | 5e−6, 1e−5, 2e−5 | 2e−4, 1e−4 | 1e−5, 2e−5, 5e−6 | 1e−4, 1e−3 |
| Adapter LR | 5e−6 | 2e−5 | 1e−6 | 1e−6 |
| Effective batch size* | 128 | 128 | 16 | 64 |
| LoRA rank | — | 8 | — | 8 |

Table 11: Hyperparameter settings for LLaVA-v1.5-7B and Qwen2-VL-7B-Instruct under standard fine-tuning and LoRA. *Effective batch size accounts for gradient accumulation across 8 GPUs.

We perform continual fine-tuning of the model weights on 8 NVIDIA A6000 GPUs. Notice Qwen2-VL models are trained with much smaller batch sizes due to heavier GPU VRAM requirements.

### A.2 MULTI-MODAL SEMANTIC PERTURBATION

Multi-modal semantic perturbation is a two-stage process. In the first process where we obtain prompts that will be provided to Flux Controlnet model is generated by using GPT-4o with `api-version=2024-08-01-preview`. We set `temperature=0.3` and limit `max_tokens=800`.

In the second stage where we create a perturbed version of the original image based on the caption generated in the first stage, we utilize Controlnet trained with Flux diffusion model [5]. We follow the default hyperparameter settings in the repository, except that we enforce the generated images to have the same resolution as the original images. All images are generated with 25 steps.

### A.3 COST OF MULTI-MODAL SEMANTIC PERTURBATION

First, we note that our semantic perturbation pipeline only needs to be run once to constructed the perturbed benchmark. We also note that the cost scales linearly with the size of the dataset. We list below the cost and hardware requirements of each stage of our pipeline.

**Caption generation.** Using GPT-4o as the captioning model required on average ∼75 input tokens and ∼125 output tokens per image. This means that generating 1,000 captions would cost less than \$1.50 (USD). This stage also does not require proprietary models: it can be replaced with Molmo-7B-D (Table 10, 23, 24), which fits on a single 24GB GPU.

**Flux+ControlNet.** Flux+ControlNet generation runs on a single 40GB GPU. With 25 sampling steps, each image takes around 15 20 seconds to generate, and this can be further reduced by decreasing the number of steps. This stage is trivially parallelizable across multiple GPUs or machines if higher throughput is needed.

**Automated filtering with o3.** Automated filtering required on average ∼75 input tokens and ∼200 output tokens per image. Generating 1,000 filtering decisions costs less than \$2.00 (USD).

---

[4]The default training script can be found here: https://github.com/haotian-liu/LLaVA.
[5]Code can be accessed here: https://github.com/XLabs-AI/x-flux

Overall, the total cost to construct the perturbed benchmark is cheap with modest hardware requirements. Our approach scales linearly with dataset size, making our approach practical and scalable even for reasonably large benchmarks. Finally, when real-world counterfactuals are already available as part of the benchmark design (e.g., NaturalBench in Table 6, 20), we do not need to generate perturbations at all.

## B  SYSTEM PROMPTS TO GPT-4O AND O3

To generate the semantically perturbed question, we utilize GPT-4o OpenAI (2024) and Flux diffusion model Labs (2024) trained on ControlNet Zhang et al. (2023). We generate a detailed caption about the image using GPT-4o *along with the original question, original answer and the new correct answer* with the following system prompt:

> *Your job is to generate a text-to-image prompt that can be used with a diffusion model. Based on the question and answer, write a detailed caption so that all necessary details are included and the question remains solvable.*
>
> *Additionally, modify the image so that the correct answer changes. For example, if the question asks "How many people are in the image?", change the image to have more people.*

In Section 7, we demonstrated that the manual filtering process can be automated with o3 model which is a powerful reasoning model. We utilize the following system prompt to generate model responses that can be used to filter the invalid perturbed images.

> *You will be given a question and an image pair, along with the answer. Your job is to critically analyze the image-question pair to verify that the question can be correctly answered.*
>
> *In particular, ensure that one can deduce the correct answer choice and that choice only. If there is any ambiguity, you must reject this question. When finalizing your decision, do*
>
> *NOT take into consideration the quality of the image. As long as the question remains solvable, you should keep it. Provide your answer in the following format: "Answer: ANSWER" and answer with KEEP or REJECT.*

## C  PERPLEXITY-BASED METHODS

We evaluate three perplexity-based methods: Shared Likelihood (SL) (Oren et al., 2023), which compares log-likelihoods on original versus shuffled inputs; Guided Prompting (GP) (Golchin & Surdeanu, 2024), which uses an LLM to score model completions against the original; and N-gram Accuracy (NG) (Xu et al., 2024), which masks answer choices and checks reproduction. Each probes whether models memorize test samples through partial or perturbed inputs in the text domain.

Results in Table 12 show that SL, GP, and NG fail to meet any of the requirements. All three depend on reference values from clean models, violating Requirement 1. Contaminated models often display counter-intuitive trends, such as improved scores with greater contamination, violating Requirement 2. Finally, none satisfy Requirement 3—for instance, SL p-values increase rather than decrease as contamination intensifies, directly contradicting expectation.

## D  DETECTING CONTAMINATED MODELS WITH MULTI-MODAL SEMANTIC PERTURBATION ON REALWORLDQA

Table 13 reports results for RealWorldQA. Similar to results on MMStar, we find that (i) clean models consistently outperform contaminated ones, confirming that perturbed questions are not harder; (ii) in contrast, contaminated models show clear performance drops, enabling reliable detection without thresholds or prior knowledge (Requirement 1). The drop scales with contamination level (Requirement 3) and holds across training strategies (Requirement 2).

| Method | Epoch | RealWorldQA | | | MMStar | | |
|---|---|---|---|---|---|---|---|
| | | SL (p-val)(↓) | GP (%)(↑) | NG (%)(↑) | SL (p-val)(↓) | GP (%)(↑) | NG (%)(↑) |
| LLaVA -v1.5-7B | – | 0.926 | 7.32 | 25.67 | 0.201 | 2.53 | 14.54 |
| LoRA | 1 | 0.927 | 8.63 | 24.94 | 0.088 | 0.60 | 5.63 |
| | 2 | 0.933 | 4.44 | 26.32 | 0.066 | 0.80 | 3.27 |
| | 3 | 0.935 | 7.19 | 25.12 | 0.072 | 0.53 | 2.92 |
| LLM +MLP | 1 | 0.928 | 4.84 | 26.09 | 0.207 | 0.73 | 12.69 |
| | 2 | 0.929 | 6.67 | 26.93 | 0.219 | 0.47 | 8.63 |
| | 3 | 0.932 | 5.75 | 26.82 | 0.238 | 0.13 | 3.12 |
| Qwen2 -VL-7B | – | 0.166 | 38.43 | 13.10 | 0.006 | 13.80 | 3.08 |
| LoRA | 1 | 0.333 | 46.27 | 13.88 | 0.009 | 3.07 | 7.67 |
| | 2 | 0.166 | 49.80 | 14.51 | 0.004 | 3.00 | 8.26 |
| | 3 | 0.323 | 47.06 | 14.54 | 0.000 | 1.00 | 7.96 |
| LLM | 1 | 0.124 | 14.64 | 15.48 | 0.005 | 1.13 | 2.47 |
| | 2 | 0.070 | 9.54 | 16.08 | 0.004 | 1.20 | 3.09 |
| | 3 | 0.165 | 11.63 | 16.97 | 0.005 | 1.00 | 7.59 |

Table 12: Perplexity-based baselines (SL = shared_likelihood, GP = guided prompting, NG = n-gram) for RealWorldQA and MMStar. MMBench was left out due to limited compute. To clarify, the models are trained with the dataset they are being evaluated on. There is no clear signal that distinguishes clean and contaminated models.

## E    EXTENDED EVALUATION ON ADDITIONAL OPEN-SOURCE AND PROPRIETARY MODELS

We further apply our pipeline to GPT-4o (OpenAI, 2024), Gemini-2.0-Flash (Gemini Team, 2024), Phi-3.5-V (Abdin et al., 2024b), and InternVL-2.5 (Chen et al., 2025). We do not contaminate these models. Although we cannot guarantee that these models have never encountered our evaluation data, we *assume* they are uncontaminated. Under this assumption, if our framework is sound, their performance should remain consistent across original and perturbed datasets. Indeed, our results in Table 14 confirm this expectation. Across all models, perturbed performance exceeds original performance, reinforcing that clean models generalize while contaminated ones fail. Notably, Phi-3.5-V exhibits the largest gain, while InternVL-2.5-8B remains relatively flat, demonstrating that our metric is robust across architectures. This consistency underscores the effectiveness of multi-modal semantic perturbation as a detection framework.

## F    FULL EXPERIMENT RESULTS

In this section, we provide the full experiment results of multi-modal semantic perturbation across four training strategies that were omitted due to limited space.

### F.1    FULL RESULTS ON LLAVA-V1.5-7B AND QWEN2-VL-7B

Table 15, 16, 17, and 18 list out all results for four varying training strategies for LLaVA-v1.5-7B and Qwen2-VL-7B , including the non-default training strategies for each model which were omitted due to limited space. Note that the results here are computed on 440 manually filtered images. Our approach detects contaminated models regardless of the training strategies and satisfies all three requirements.

### F.2    FULL RESULTS WITH LLAVA-V1.5-13B AND NATURALBENCH

Table 19 lists full results for LLaVA-v1.5-13B model trained on RealWorldQA. Note that the results here are computed on 440 manually filtered images. Table 20 lists full results for LLaVA-v1.5-7B trained on one variant of NaturalBench and tested on the counterfactual version. The clean and consistent detection of contaminated models show that our approach can be applied to models of larger scale and to real-world perturbations.

| Method | Metric | LLaVA-v1.5-7B (clean) | LoRA (contaminated) | | | LLM+MLP (contaminated) | | | Require clean model? |
|---|---|---|---|---|---|---|---|---|---|
| | | — | Epoch 1 | Epoch 2 | Epoch 3 | Epoch 1 | Epoch 2 | Epoch 3 | (Practicality (Req. 1)) |
| Ours | RQA | 52.05 | 62.73 | 70.00 | 79.55 | 56.36 | 64.55 | 70.68 | |
| | RQA_P | 56.36 | 51.36 | 52.73 | 44.77 | 52.95 | 50.45 | 51.82 | No ✓ |
| | Δ | +4.31 | −11.37 | −17.27 | −34.78 | −3.41 | −14.10 | −18.86 | |
| | Success? | ✓ | ✓ | ✓ | ✓ | ✓ | ✓ | ✓ | |
| CircularEval | RQA | 52.05 | 62.73 | 70.00 | 79.55 | 56.36 | 64.55 | 70.68 | |
| | RQA_C | 36.59 | 17.95 | 45.23 | 57.50 | 33.18 | 37.05 | 43.40 | Yes ✗ |
| | Δ | -15.46 | -44.78 | -24.77 | -22.05 | -23.18 | -27.50 | -27.28 | |
| | Success? | – | ✓ | ✓ | ✓ | ✓ | ✓ | ✓ | |
| Choice Confusion | RQA | 52.05 | 62.73 | 70.00 | 79.55 | 56.36 | 64.55 | 70.68 | |
| | RQA_G | 66.59 | 41.36 | 59.09 | 66.14 | 62.95 | 62.50 | 63.64 | No ✓ |
| | Δ | +14.54 | -21.37 | -10.91 | -13.41 | +6.59 | -2.05 | -7.04 | |
| | Success? | ✓ | ✓ | ✓ | ✓ | ✗ | ✓ | ✓ | |
| Multi-modal Leakage | RQA_to | 34.77 | 37.27 | 47.05 | 48.41 | 37.73 | 40.68 | 45.00 | |
| | Δ | – | +2.50 | +12.28 | +13.64 | +2.96 | +5.91 | +10.23 | Yes ✗ |
| | Success? | – | ✓ | ✓ | ✓ | ✓ | ✓ | ✓ | |

| Method | Metric | Qwen2-VL-7B (clean) | LoRA (contaminated) | | | LLM only (contaminated) | | | Require clean model? |
|---|---|---|---|---|---|---|---|---|---|
| | | — | Epoch 1 | Epoch 2 | Epoch 3 | Epoch 1 | Epoch 2 | Epoch 3 | (Practicality (Req. 1)) |
| Ours | RQA | 70.45 | 79.32 | 87.50 | 88.86 | 74.77 | 78.64 | 85.23 | |
| | RQA_P | 71.36 | 65.00 | 64.55 | 61.59 | 46.82 | 50.45 | 46.59 | No ✓ |
| | Δ | +0.91 | −14.32 | −22.95 | −27.27 | −27.95 | −28.19 | −38.64 | |
| | Success? | ✓ | ✓ | ✓ | ✓ | ✓ | ✓ | ✓ | |
| CircularEval | RQA | 70.45 | 79.32 | 87.50 | 88.86 | 74.77 | 78.64 | 85.23 | |
| | RQA_C | 63.64 | 65.00 | 65.91 | 67.05 | 65.23 | 66.59 | 77.73 | Yes ✗ |
| | Δ | -6.81 | -14.32 | -21.59 | -21.81 | -9.54 | -12.05 | -7.50 | |
| | Success? | – | ✓ | ✓ | ✓ | ✓ | ✓ | ✓ | |
| Choice Confusion | RQA | 70.45 | 79.32 | 87.50 | 88.86 | 74.77 | 78.64 | 85.23 | |
| | RQA_G | 91.36 | 91.59 | 92.05 | 92.50 | 86.14 | 89.77 | 95.23 | No ✓ |
| | Δ | +20.91 | +12.27 | +4.55 | +3.64 | +11.37 | +11.13 | +10.00 | |
| | Success? | ✓ | ✗ | ✗ | ✗ | ✗ | ✗ | ✗ | |
| Multi-modal Leakage | RQA_to | 36.14 | 37.27 | 41.14 | 42.27 | 52.73 | 28.41 | 62.50 | |
| | Δ | – | +1.13 | +5.00 | +6.13 | +16.59 | -7.73 | +26.36 | Yes ✗ |
| | Success? | – | ✓ | ✓ | ✓ | ✓ | ✗ | ✓ | |

Table 13: Performance of LLaVA-v1.5-7B (top) and Qwen2-VL-7B (bottom) on the RealWorldQA dataset. We compare to "Multi-modal Leakage" Chen et al. (2024a), CircularEval (Liu et al., 2024b), and "Choice Confusion" Yao et al. (2024). Clean models perform better – confirming that the perturbed questions are indeed of equal or lower difficulty. In contrast, all contaminated models perform worse, enabling reliable detection by our method via a simple check for performance drops. RQA denotes RealWorldQA and "_P" denotes the semantically perturbed version; "to" denotes text-only performance; "_C" denotes evaluation using circular options; "_G" denotes evaluation using choice confusion; Δ denotes the difference in performance, with positive values indicating gains. "Success?" indicates whether the method detected contamination. "Require clean model?" indicates whether the method requires access to a clean model as a baseline. If a clean model is required as reference, the method cannot be used to detect the reference models themselves, so the entry is marked as "–". For full results, refer to Appendix F.

| Model | RQA | RQA_P | Δ |
|---|---|---|---|
| Gemini-2.0-Flash | 68.37 | 71.59 | +3.22 |
| GPT-4o | 65.68 | 69.93 | +4.25 |
| Phi3.5-Vision | 52.68 | 68.86 | +16.18 |
| InternVL-2.5-8B | 64.05 | 64.77 | +0.72 |

Table 14: Accuracy on the original vs. perturbed datasets for open-source and proprietary models. Δ indicates the accuracy change.

### F.3 FULL RESULTS WITH ○3 FILTERING

Table 21 and 22 list out all results for contamination detection results after filtering the perturbed images with ○3 model. To clarify, filtering with ○3 model results in 294 images. Our approach still detects contaminated models and satisfies all three requirements, proving that our pipeline design is modular and can be automated.

| Model | Epoch | RQA (%) | RQA_P (%) | Δ |
|---|---|---|---|---|
| **LLaVA-v1.5-7B** | — | 52.05 | 56.36 | +4.31 |
| **Contamination with RealWorldQA** | | | | |
| **LoRA** | 1 | 62.73 | 51.36 | −11.37 |
| | 2 | 70.00 | 52.73 | −17.27 |
| | 3 | 79.55 | 44.77 | −34.78 |
| **LLM only** | 1 | 58.41 | 51.59 | −6.82 |
| | 2 | 63.64 | 50.68 | −12.96 |
| | 3 | 70.91 | 47.05 | −23.86 |
| **LLM+MLP** | 1 | 56.36 | 52.95 | −3.41 |
| | 2 | 64.55 | 50.45 | −14.10 |
| | 3 | 70.68 | 51.82 | −18.86 |
| **ALL** | 1 | 56.36 | 54.32 | −2.04 |
| | 2 | 64.77 | 52.73 | −12.04 |
| | 3 | 70.23 | 52.05 | −18.18 |

Table 15: Performance of clean and contaminated LLaVA-v1.5-7B models on RealWorldQA. "_P" denotes the semantically perturbed version. All three requirements are satisfied.

| Model | Epoch | RQA (%) | RQA_P (%) | Δ |
|---|---|---|---|---|
| **Qwen2-VL-7B** | — | 70.45 | 71.36 | +0.91 |
| **Contamination with RealWorldQA** | | | | |
| **LoRA** | 1 | 79.32 | 65.00 | −14.32 |
| | 2 | 87.50 | 64.55 | −22.95 |
| | 3 | 88.86 | 61.59 | −27.27 |
| **LLM only** | 1 | 74.77 | 46.82 | −27.95 |
| | 2 | 78.64 | 50.45 | −28.19 |
| | 3 | 85.23 | 46.59 | −38.64 |
| **LLM+MLP** | 1 | 75.23 | 57.95 | −17.28 |
| | 2 | 88.18 | 49.77 | −38.41 |
| | 3 | 93.18 | 47.50 | −45.68 |
| **ALL** | 1 | 74.77 | 39.09 | −35.68 |
| | 2 | 78.18 | 45.91 | −32.27 |
| | 3 | 87.50 | 40.45 | −47.05 |

Table 16: Performance of clean and contaminated Qwen2-VL-7B models on RealWorldQA. "_P" denotes the semantically perturbed version. All three requirements are satisfied.

| Model | Epoch | RQA (%) | RQA_P (%) | Δ |
|---|---|---|---|---|
| **LLaVA-v1.5-7B** | — | 37.78 | 69.29 | +31.51 |
| **Contamination with MMStar** | | | | |
| **LoRA** | 1 | 52.53 | 44.24 | −8.29 |
| | 2 | 50.71 | 37.58 | −13.13 |
| | 3 | 54.34 | 38.18 | −16.16 |
| **LLM only** | 1 | 44.85 | 37.98 | −6.87 |
| | 2 | 48.48 | 38.99 | −9.49 |
| | 3 | 55.15 | 38.59 | −16.56 |
| **LLM+MLP** | 1 | 39.39 | 32.12 | −7.27 |
| | 2 | 49.70 | 38.38 | −11.32 |
| | 3 | 53.54 | 38.99 | −14.55 |
| **ALL** | 1 | 41.82 | 33.33 | −8.49 |
| | 2 | 48.89 | 37.37 | −11.52 |
| | 3 | 50.71 | 36.97 | −13.74 |

Table 17: Performance of clean and contaminated LLaVA-v1.5-7B models on MMStar. "_P" denotes the semantically perturbed version. All three requirements are satisfied.

| Method | Epoch | RQA (%) | RQA_P (%) | Δ |
|---|---|---|---|---|
| **Qwen2-VL-7B** | — | 62.02 | 78.18 | +16.16 |
| **Contamination with MMStar** | | | | |
| **LoRA** | 1 | 77.37 | 73.33 | −4.04 |
| | 2 | 87.88 | 68.48 | −19.40 |
| | 3 | 91.52 | 67.47 | −24.05 |
| **LLM only** | 1 | 80.20 | 50.71 | −29.49 |
| | 2 | 94.95 | 52.32 | −42.63 |
| | 3 | 97.17 | 49.09 | −48.08 |
| **LLM+MLP** | 1 | 83.43 | 55.96 | −27.47 |
| | 2 | 94.95 | 52.93 | −42.02 |
| | 3 | 97.98 | 51.11 | −46.87 |
| **ALL** | 1 | 71.31 | 47.27 | −24.04 |
| | 2 | 93.13 | 43.64 | −49.49 |
| | 3 | 96.77 | 44.04 | −52.73 |

Table 18: Performance of clean and contaminated Qwen2-VL-7B models on MMStar. "_P" denotes the semantically perturbed version. All three requirements are satisfied.

### F.4 FULL RESULTS WITH MOLMO-7B-D AS CAPTIONING MODEL

Table 23 and 24 list results for LLaVA-v1.5-7B and Qwen2-VL-7B evaluated on perturbed images generated from Molmo-7B-D captions. Note that this pipeline yields 398 valid image-question pairs after manual filtering. Both results show clean detection trends, satisfying all requirements.

| Model | Epoch | RQA (%) | RQA_P (%) | Δ |
|---|---|---|---|---|
| **LLaVA-v1.5-13B** | — | 51.14 | 57.27 | +6.13 |
| **Contamination with RealWorldQA** | | | | |
| **LoRA** | 1 | 74.32 | 38.18 | −36.14 |
| | 2 | 73.18 | 32.73 | −40.45 |
| | 3 | 77.05 | 34.77 | −42.28 |
| **LLM only** | 1 | 57.05 | 44.77 | −12.28 |
| | 2 | 68.18 | 44.32 | −23.86 |
| | 3 | 68.64 | 43.86 | −24.78 |
| **LLM+MLP** | 1 | 56.59 | 37.50 | −19.09 |
| | 2 | 71.59 | 38.86 | −32.73 |
| | 3 | 75.45 | 37.27 | −38.18 |
| **ALL** | 1 | 57.73 | 44.55 | −13.18 |
| | 2 | 68.41 | 44.32 | −24.09 |
| | 3 | 69.09 | 43.41 | −25.68 |

Table 19: Performance of LLaVA-v1.5-13B models on the RealWorldQA benchmark and its semanti- cally perturbed version consisting of 440 image-question pairs. All three requirments are satisfied.

| Model | Epoch | Train (%) | Test (%) | Δ |
|---|---|---|---|---|
| **LLaVA-v1.5-7B** | — | 65.63 | 65.89 | +0.26 |
| **Contamination with NaturalBench** | | | | |
| **LoRA** | 1 | 81.53 | 61.37 | −20.16 |
| | 2 | 89.79 | 57.16 | −32.63 |
| | 3 | 91.11 | 57.32 | −33.79 |
| **LLM only** | 1 | 77.63 | 61.95 | −15.68 |
| | 2 | 88.11 | 57.89 | −30.21 |
| | 3 | 90.58 | 57.32 | −33.26 |
| **LLM+MLP** | 1 | 79.95 | 58.79 | −21.16 |
| | 2 | 97.05 | 54.32 | −42.74 |
| | 3 | 98.63 | 53.05 | −45.58 |
| **ALL** | 1 | 81.84 | 58.42 | −23.42 |
| | 2 | 97.05 | 54.47 | −42.58 |
| | 3 | 98.63 | 52.68 | −45.95 |

Table 20: Performance of clean and contaminated LLaVA-v1.5-7B models on Natural-Bench. Test set denotes a natural counterfactual version of the train set. Clean model maintains a similar performance while contaminated models drop in performance for upto -45.95%.

| Model | Epoch | RQA (%) | RQA_P (%) | Δ |
|---|---|---|---|---|
| **Contamination with RealWorldQA** | | | | |
| **LLaVA (clean)** | — | 50.68 | 59.86 | +9.18 |
| **LoRA (contaminated)** | 1 | 50.34 | 44.90 | −5.44 |
| | 2 | 66.33 | 46.60 | −19.73 |
| | 3 | 77.21 | 47.28 | −29.93 |
| **LLM (contaminated)** | 1 | 59.86 | 56.80 | −3.06 |
| | 2 | 63.95 | 55.10 | −8.85 |
| | 3 | 68.71 | 50.00 | −18.71 |
| **LLM+MLP (contaminated)** | 1 | 58.50 | 54.42 | −4.08 |
| | 2 | 60.54 | 54.42 | −6.12 |
| | 3 | 64.97 | 53.40 | −11.57 |
| **ALL (contaminated)** | 1 | 59.86 | 56.46 | −3.40 |
| | 2 | 64.63 | 53.40 | −11.23 |
| | 3 | 69.05 | 50.34 | −18.71 |

Table 21: Performance of clean and contaminated LLaVA-v1.5-7B models on RealWorldQA. "_P" denotes the semantically perturbed version. Note that accuracies are measured on 294 filtered images. All three requirements are satisfied.

| Model | Epoch | RQA (%) | RQA_P (%) | Δ |
|---|---|---|---|---|
| **Contamination with RealWorldQA** | | | | |
| **Qwen2-VL (clean)** | — | 74.15 | 74.83 | +0.68 |
| **LoRA (contaminated)** | 1 | 77.21 | 74.83 | −2.38 |
| | 2 | 78.91 | 74.15 | −4.76 |
| | 3 | 79.59 | 74.15 | −5.44 |
| **LLM (contaminated)** | 1 | 79.25 | 56.12 | −23.13 |
| | 2 | 85.37 | 54.42 | −30.95 |
| | 3 | 90.48 | 55.44 | −35.04 |
| **LLM+MLP (contaminated)** | 1 | 77.89 | 62.24 | −15.65 |
| | 2 | 87.41 | 57.48 | −29.93 |
| | 3 | 88.78 | 58.16 | −30.62 |
| **ALL (contaminated)** | 1 | 80.27 | 58.50 | −21.77 |
| | 2 | 90.82 | 53.40 | −37.42 |
| | 3 | 92.52 | 52.04 | −40.48 |

Table 22: Performance of clean and contaminated Qwen2-VL-7B models on RealWorldQA. "_P" denotes the semantically perturbed version. Note that accuracies are measured on 294 filtered images. All three requirements are satisfied.

## F.5 FULL RESULTS WITH CONTAMINATION WITH PARAPHRASED DATA

Table 25 and 26 list full results for LLaVA-v1.5-7B and Qwen2-VL-7B models on RealWorldQA and table 27 and 28 on MMStar. To clarify, the models are trained with a paraphrased version of the original benchmark, and are evaluated on the original benchmark.

More concretely, we prompt GPT-4o to generate three possible paraphrases of the original question, and select the one with the lowest 5-gram overlap with the original question, enforcing the model to use different sentence structures and vocabularies.

Furthermore, we observe that models contaminated on the paraphrased version of the questions show inflated performance on the original benchmark, validating our paraphrasing pipeline. The clean and consistent detection proves that our perturbation pipeline is robust against paraphrasing, which is a challenging setting for contamination detection.

| Model | Config | RQA (%) | RQA_P (%) | Δ (%) |
|---|---|---|---|---|
| **Contamination with RealWorldQA** | | | | |
| **LLaVA (clean)** | – | 44.22 | 52.01 | +7.79 |
| **LoRA (contaminated)** | 1 | 44.97 | 35.93 | −9.04 |
| | 2 | 55.78 | 32.66 | −23.12 |
| | 3 | 70.35 | 32.91 | −37.44 |
| **LLM (contaminated)** | 1 | 52.76 | 41.96 | −10.80 |
| | 2 | 53.02 | 43.97 | −9.05 |
| | 3 | 53.52 | 43.72 | −9.80 |
| **LLM+MLP (contaminated)** | 1 | 56.53 | 39.95 | −16.58 |
| | 2 | 60.80 | 39.45 | −21.35 |
| | 3 | 61.56 | 39.95 | −21.61 |
| **ALL (contaminated)** | 1 | 62.81 | 35.43 | −27.38 |
| | 2 | 61.81 | 35.93 | −25.88 |
| | 3 | 62.56 | 35.68 | −26.88 |

Table 23: Performance of clean and contaminated LLaVA-v1.5-7B models on RealWorldQA. "_P" denotes the semantically perturbed version. Note that accuracies are measured on 398 filtered images. All three requirements are satisfied.

| Model | Config | RQA (%) | RQA_P (%) | Δ (%) |
|---|---|---|---|---|
| **Contamination with RealWorldQA** | | | | |
| **Qwen2-VL (clean)** | – | 61.81 | 65.08 | +3.27 |
| **LoRA (contaminated)** | 1 | 63.57 | 59.05 | −4.52 |
| | 2 | 67.84 | 52.26 | −15.58 |
| | 3 | 68.84 | 52.26 | −16.58 |
| **LLM (contaminated)** | 1 | 78.14 | 40.70 | −37.44 |
| | 2 | 84.67 | 32.16 | −52.51 |
| | 3 | 87.19 | 35.43 | −51.76 |
| **LLM+MLP (contaminated)** | 1 | 73.12 | 45.48 | −27.64 |
| | 2 | 83.42 | 34.67 | −48.75 |
| | 3 | 85.18 | 35.93 | −49.25 |
| **ALL (contaminated)** | 1 | 76.38 | 39.95 | −36.43 |
| | 2 | 88.94 | 32.41 | −56.53 |
| | 3 | 90.20 | 31.41 | −58.79 |

Table 24: Performance of clean and contaminated Qwen2-VL-7B models on RealWorldQA. "_P" denotes the semantically perturbed version. Note that accuracies are measured on 398 filtered images. All three requirements are satisfied.

| Model | Epoch | RQA (%) | RQA_P (%) | Δ |
|---|---|---|---|---|
| **Contamination with paraphrased RealWorldQA** | | | | |
| **LLaVA (clean)** | — | 52.05 | 56.36 | +4.31 |
| **LoRA (contaminated)** | 1 | 52.03 | 40.00 | -12.03 |
| | 2 | 55.91 | 40.00 | -15.91 |
| | 3 | 59.09 | 38.86 | -20.23 |
| **LLM (contaminated)** | 1 | 56.14 | 54.09 | -2.05 |
| | 2 | 60.91 | 49.09 | -11.82 |
| | 3 | 63.64 | 50.45 | -13.19 |
| **LLM+MLP (contaminated)** | 1 | 56.14 | 53.41 | -2.73 |
| | 2 | 61.36 | 48.64 | -12.72 |
| | 3 | 63.64 | 49.77 | -13.87 |
| **ALL (contaminated)** | 1 | 55.91 | 53.64 | -2.27 |
| | 2 | 62.05 | 48.86 | -13.19 |
| | 3 | 63.86 | 50.00 | -13.86 |

Table 25: Performance of clean and contaminated LLaVA-v1.5-7B models on RealWorldQA. "_P" denotes the semantically perturbed version. Note that the models are trained with paraphrased version of the original benchmark. All three requirements are satisfied.

| Model | Epoch | RQA (%) | RQA_P (%) | Δ |
|---|---|---|---|---|
| **Contamination with paraphrased RealWorldQA** | | | | |
| **Qwen2-VL (clean)** | — | 70.45 | 71.59 | +1.14 |
| **LoRA (contaminated)** | 1 | 72.50 | 71.59 | -0.91 |
| | 2 | 73.86 | 69.55 | -4.31 |
| | 3 | 75.23 | 69.32 | -5.91 |
| **LLM (contaminated)** | 1 | 78.41 | 53.86 | -24.55 |
| | 2 | 87.05 | 56.59 | -30.46 |
| | 3 | 89.55 | 54.32 | -35.23 |
| **LLM+MLP (contaminated)** | 1 | 80.00 | 52.50 | -27.50 |
| | 2 | 87.05 | 56.59 | -30.46 |
| | 3 | 88.64 | 55.23 | -33.41 |
| **ALL (contaminated)** | 1 | 75.45 | 51.82 | -23.63 |
| | 2 | 86.36 | 56.36 | -30.00 |
| | 3 | 90.23 | 55.45 | -34.78 |

Table 26: Performance of clean and contaminated Qwen2-VL-7B models on RealWorldQA. "_P" denotes the semantically perturbed version. Note that the models are trained with paraphrased version of the original benchmark. All three requirements are satisfied.

## F.6 FULL RESULTS WITH CONTAMINATION WITH A MIXTURE OF DATASETS

Table 29 and 30 list full results for LLaVA-v1.5-7B and Qwen2-VL-7B models on RealWorldQA and table 31 and 32 on MMStar. All models have been fine-tuned on a mixture of 6 popular multi-modal benchmarks: MathVista Lu et al. (2024b), MMBench Liu et al. (2024b), MMMU Yue et al. (2024), CV-Bench Tong et al. (2024), MMStar and RealWorldQA, resulting in 11,280 image-question pairs. We test whether our approach can still detect contamination on RealWorldQA and MMStar, which now consist only of ∼6.7% and ∼13.3% of the fine-tuning dataset, respectively. All four tables show clean and consistent detection results, satisfying all three requirements.

| Model | Epoch | MMStar (%) | MMStar_P (%) | Δ |
|---|---|---|---|---|
| **Contamination with paraphrased MMStar** | | | | |
| **LLaVA (clean)** | — | 37.78 | 68.29 | +31.51 |
| **LoRA (contaminated)** | 1 | 51.52 | 43.64 | -7.88 |
| | 2 | 59.60 | 41.82 | -17.78 |
| | 3 | 62.22 | 40.81 | -21.41 |
| **LLM (contaminated)** | 1 | 47.47 | 45.68 | -1.79 |
| | 2 | 53.13 | 50.39 | -2.74 |
| | 3 | 61.82 | 53.37 | -8.45 |
| **LLM+MLP (contaminated)** | 1 | 47.27 | 46.06 | -1.13 |
| | 2 | 57.60 | 53.54 | -3.06 |
| | 3 | 62.42 | 57.98 | -4.44 |
| **ALL (contaminated)** | 1 | 46.87 | 45.25 | -1.62 |
| | 2 | 53.33 | 49.52 | -3.81 |
| | 3 | 62.42 | 57.58 | -4.84 |

Table 27: Performance of clean and contaminated LLaVA-v1.5-7B models on MMStar. "_P" denotes the semantically perturbed version. Note that the models are trained with paraphrased version of the original benchmark. All three requirements are satisfied.

| Model | Epoch | MMStar (%) | MMStar_P (%) | Δ |
|---|---|---|---|---|
| **Contamination with paraphrased MMStar** | | | | |
| **Qwen2-VL (clean)** | — | 62.02 | 78.18 | +16.16 |
| **LoRA (contaminated)** | 1 | 78.87 | 67.48 | -11.39 |
| | 2 | 83.33 | 64.95 | -18.38 |
| | 3 | 85.70 | 63.74 | -21.96 |
| **LLM (contaminated)** | 1 | 89.09 | 60.61 | -28.48 |
| | 2 | 96.36 | 55.76 | -40.60 |
| | 3 | 97.98 | 54.34 | -43.64 |
| **LLM+MLP (contaminated)** | 1 | 87.27 | 61.62 | -25.65 |
| | 2 | 97.17 | 55.96 | -41.21 |
| | 3 | 97.78 | 56.36 | -41.42 |
| **ALL (contaminated)** | 1 | 80.20 | 63.64 | -16.56 |
| | 2 | 94.95 | 54.95 | -40.00 |
| | 3 | 96.77 | 55.15 | -41.62 |

Table 28: Performance of clean and contaminated Qwen2-VL-7B models on MMStar. "_P" denotes the semantically perturbed version. Note that the models are trained with paraphrased version of the original benchmark. All three requirements are satisfied.

| Model | Epoch | RQA (%) | RQA_P (%) | Δ |
|---|---|---|---|---|
| **Contamination with mixture of 6 benchmarks** | | | | |
| **LLaVA (clean)** | — | 52.05 | 56.36 | +4.31 |
| **LoRA (contaminated)** | 1 | 45.91 | 41.59 | -4.32 |
| | 2 | 62.73 | 40.23 | -22.50 |
| | 3 | 75.45 | 42.05 | -33.40 |
| **LLM (contaminated)** | 1 | 59.55 | 52.27 | -7.28 |
| | 2 | 65.68 | 50.23 | -15.45 |
| | 3 | 67.05 | 46.14 | -20.91 |
| **LLM+MLP (contaminated)** | 1 | 57.95 | 52.05 | -5.90 |
| | 2 | 61.14 | 50.91 | -10.23 |
| | 3 | 64.32 | 48.86 | -15.46 |
| **ALL (contaminated)** | 1 | 60.00 | 51.82 | -8.18 |
| | 2 | 65.91 | 48.86 | -17.05 |
| | 3 | 67.95 | 46.82 | -21.13 |

Table 29: Performance of clean and contaminated LLaVA-v1.5-7B models on RealWorldQA. "_P" denotes the semantically perturbed version. All models are contaminated with a mixture of 6 benchmarks, resulting in 11,280 image-question pairs.

| Model | Epoch | RQA (%) | RQA_P (%) | Δ |
|---|---|---|---|---|
| **Contamination with mixture of 6 benchmarks** | | | | |
| **Qwen2-VL (clean)** | — | 70.45 | 71.36 | +0.91 |
| **LoRA (contaminated)** | 1 | 72.95 | 70.00 | -2.95 |
| | 2 | 75.91 | 68.18 | -7.73 |
| | 3 | 76.14 | 68.31 | -7.83 |
| **LLM (contaminated)** | 1 | 77.50 | 50.00 | -27.50 |
| | 2 | 85.00 | 45.68 | -39.32 |
| | 3 | 88.18 | 47.95 | -40.23 |
| **LLM+MLP (contaminated)** | 1 | 75.45 | 53.41 | -22.04 |
| | 2 | 85.45 | 48.18 | -37.27 |
| | 3 | 86.36 | 47.50 | -38.86 |
| **ALL (contaminated)** | 1 | 78.64 | 50.00 | -28.64 |
| | 2 | 87.73 | 46.59 | -41.14 |
| | 3 | 89.32 | 46.59 | -42.73 |

Table 30: Performance of clean and contaminated Qwen2-VL-7B models on RealWorldQA. "_P" denotes the semantically perturbed version. All models are contaminated with a mixture of 6 benchmarks, resulting in 11,280 image-question pairs.

| Model | Epoch | MMStar (%) | MMStar_P (%) | Δ |
|---|---|---|---|---|
| **Contamination with mixture of 6 benchmarks** | | | | |
| **LLaVA (clean)** | — | 37.78 | 68.29 | +31.51 |
| **LoRA (contaminated)** | 1 | 45.86 | 26.67 | -19.19 |
| | 2 | 83.43 | 54.14 | -29.29 |
| | 3 | 85.25 | 52.93 | -32.32 |
| **LLM (contaminated)** | 1 | 42.73 | 38.28 | -4.45 |
| | 2 | 62.02 | 35.15 | -26.87 |
| | 3 | 73.54 | 41.82 | -31.72 |
| **LLM+MLP (contaminated)** | 1 | 34.65 | 33.03 | -1.62 |
| | 2 | 62.42 | 38.59 | -23.83 |
| | 3 | 82.22 | 48.48 | -33.74 |
| **ALL (contaminated)** | 1 | 32.22 | 30.41 | -1.81 |
| | 2 | 77.78 | 45.05 | -32.73 |
| | 3 | 81.82 | 45.45 | -36.37 |

Table 31: Performance of clean and contaminated LLaVA-v1.5-7B models on MMStar. "_P" denotes the semantically perturbed version. All models are contaminated with a mixture of 6 benchmarks, resulting in 11,280 image-question pairs.

| Model | Epoch | MMStar (%) | MMStar_P (%) | Δ |
|---|---|---|---|---|
| **Contamination with mixture of 6 benchmarks** | | | | |
| **Qwen2-VL (clean)** | — | 62.02 | 78.18 | +16.16 |
| **LoRA (contaminated)** | 1 | 78.38 | 71.31 | -7.07 |
| | 2 | 94.14 | 65.25 | -28.89 |
| | 3 | 95.96 | 63.64 | -32.32 |
| **LLM (contaminated)** | 1 | 89.90 | 60.40 | -29.50 |
| | 2 | 97.98 | 54.95 | -43.03 |
| | 3 | 98.99 | 55.96 | -43.03 |
| **LLM+MLP (contaminated)** | 1 | 90.10 | 61.41 | -28.69 |
| | 2 | 97.17 | 55.56 | -41.61 |
| | 3 | 98.79 | 55.96 | -42.83 |
| **ALL (contaminated)** | 1 | 85.86 | 62.83 | -23.03 |
| | 2 | 89.70 | 57.37 | -32.33 |
| | 3 | 93.74 | 55.76 | -37.98 |

Table 32: Performance of clean and contaminated Qwen2-VL-7B models on RealWorldQA. "_P" denotes the semantically perturbed version. All models are contaminated with a mixture of 6 benchmarks, resulting in 11,280 image-question pairs.

## F.7 FULL RESULTS WITH CONTAMINATION IN FREE-FORM QA TASK

To verify that our perturbation pipeline extends beyond multiple-choice VQA tasks, we use Counter-Curate Zhang et al. (2024a), which consists of synthetically generated counterfactual images designed to test compositional knowledge. More concretely, CounterCurate contains a subset of images that are generated from DALLE-3 with prompts that only differ in the attributes or noun (e.g. "A woman with blue hat" vs. "A man with red hat"). Hence both images remain relatively consistent, semantically and spatially.

While CounterCurate is originally a multiple-choice VQA benchmark, the answer options are natural sentences, making it suitable for testing model's free-form QA. We reformulate the task into a captioning setup: we ask the model to generate an overview description of the image focusing on the main entity and treat the correct choice as the ground-truth caption. We then use GPT-4o as a judge to decide whether the model's description is closer to the ground-truth caption or to the caption of the counterfactual image.

Concretely, we randomly sample 500 image-question pairs from the dataset and use the corresponding 500 counterfactuals as the perturbed test set. In this setting we cannot expect the clean model to perform better on the counterfactual set as the counterfactuals need not be easier than the original; instead, we look at how the performance gap between original vs. counterfactual changes under contamination.

Since we are testing the models on a free-form language generation task with an unlimited output space, evaluated by an LLM-as-a-judge rather than exact matching over a fixed set of options, the contamination signal can be weaker. As a result, we see some failure cases for models trained for 1 epoch, with standard fine-tuning. However, our pipeline still detects all other contaminated models and contaminated models show a larger performance degradation between the original and counterfactual sets than the clean model, indicating that the core principle of our pipeline extends to free-form setup beyond multiple-choice VQA.

| Model | Epoch | Train (%) | Test (%) | Δ |
|---|---|---|---|---|
| **LLaVA** | — | 72.20 | 76.87 | +4.68 |
| **LoRA** | 1 | 89.44 | 88.65 | -0.79 |
| | 2 | 89.66 | 88.44 | -1.22 |
| | 3 | 90.30 | 87.79 | -2.51 |
| **LLM** | 1 | 87.72 | 88.87 | +1.15 |
| | 2 | 88.79 | 86.72 | -2.07 |
| | 3 | 90.73 | 86.94 | -3.79 |
| **LLM+MLP** | 1 | 87.72 | 88.01 | +0.29 |
| | 2 | 89.44 | 86.08 | -3.36 |
| | 3 | 89.87 | 85.84 | -4.03 |
| **ALL** | 1 | 87.72 | 88.87 | +1.15 |
| | 2 | 90.09 | 86.94 | -3.15 |
| | 3 | 90.30 | 85.87 | -4.43 |

Table 33: Performance of clean and contaminated models on the training and test splits of CounterCurate. The train set denotes a free-form QA task generated form a random sample of 500 image-caption pairs, and the test set denotes the counterfactuals. Red color denotes cases where our method will fail to detect contamination. Accuracies are measured with an LLM-as-a-judge. Since this is a free-form langauge generation task, contamination signals are weaker, leading to some detection failures under light contamination (standard fine-tuning for 1 epoch with LLM, LLM+MLP, ALL), while these settings become detectable when contamination is stronger. Across all configurations, contaminated models still show a larger degradation between the original and counterfactual sets than the clean model, indicating that the core principle of our pipeline extends to the free-form setup.

