# OpenReview forum: "Contamination Detection for VLMs Using Multi‑Modal Semantic Perturbations"
_ICLR.cc/2026/Conference — ICLR 2026 Poster_

### Official Review · Reviewer_6vCV · 2025-10-30

**Soundness:** 3
**Presentation:** 3
**Contribution:** 3
**Rating:** 6
**Confidence:** 4

**Summary:**

This paper tackles the underexplored problem of detecting test-set contamination in Vision–Language Models (VLMs) — as opposed to merely avoiding it via decontamination. Instead of text‐only perturbations used for LLMs, the authors propose a multi-modal semantic perturbation pipeline that manipulates the visual scene while preserving composition, thus minimally shifting input semantics yet changing the ground-truth answer. They contaminate LLaVA and Qwen2-VL under controlled fine-tuning regimes and show that existing detection baselines (e.g., shared likelihood, guided prompting, circular eval, choice confusion) fail to reliably track contamination. Their method exhibits high practicality (black-box only), reliability across fine-tuning types, and consistent monotonic signal w.r.t. contamination degree. They further validate on natural counterfactuals (NaturalBench), larger models (LLaVA-13B), and even simulated pretraining leakage, demonstrating generality.

**Strengths:**

The paper poses a non-generic, well-defined, and deep problem — not merely “robustness” or “performance drops,” but specifically how to detect contamination without assuming prior access to clean references or to the training corpus. The authors not only identify but formalize the essential requirements (practicality, reliability, consistency) and then prove why existing methods violate them, rather than merely benchmarking blindly. The methodology is elegant but grounded — the perturbation pipeline is semantically meaningful, not superficial image corruption — and their experimental design is unusually thorough and fair (multiple contamination strategies, ablations, real counterfactuals, automated filtering, alternative captioners).

**Weaknesses:**

The approach implicitly depends on the availability of strong controlled semantic editors (Flux + GPT-4o + ControlNet); although ablations with Molmo and automated filtering are shown, the method’s feasibility still assumes future generative tools remain capable and unbiased.

The evaluation domain is limited to visual-grounded multiple-choice VQA benchmarks (e.g., RealWorldQA, MMStar); it is argued that free-form QA is possible, but no concrete evidence is provided.

While the method is claimed fully black-box, it still rests on the assumption that perturbed samples are truly non-harder than originals: a subtle but critical assumption, mainly supported indirectly via model behavior rather than formal difficulty guarantees.

**Questions:**

Your framework assumes that the perturbed version is of comparable or lower difficulty than the original, but this is only inferred indirectly via clean model behavior. How do you enforce or guarantee this assumption beyond empirical observation? Could there exist cases where the perturbation unintentionally increases difficulty and generates false positives?

Your method relies on having access to a generative model strong enough to produce controlled, faithful semantic perturbations. In lower-resource or restricted deployment regimes, do you still consider your method “practical” (Req. 1)? What is your definition of practicality beyond using “a black box”?

---

> ### Author Response · Authors · 2025-11-23
>
> We sincerely thank Reviewer 6vCV for recognizing the **non-generic, well-defined and deep problem** we are trying to solve, and the **elegant methodology and thorough experiments** we have conducted. The reviewer’s main concerns are about the practicality and soundness of our approach, which we address below.
>
> **(W1) Capability of generative models**:
> - While our main pipeline consists of generating perturbed image-question pairs with generative models, we have shown with NaturalBench dataset (Tab. 4) that when real-world counterfactual image-question pairs exist, these can be used in the same manner to detect contamination.
> - Also, we believe it is reasonable to assume that the model capabilities will remain as capable, if not better in the future.
>
> **(W2) Free-form QA**
> - Thank you for raising this point. To provide concrete evidence beyond multiple-choice VQA, we run an additional experiment on CounterCurate[1], which consists of synthetically generated counterfactual images designed to test compositional knowledge.
> - More concretely, CounterCurate contains a subset of images that are generated from DALLE-3 with prompts that only differ in the attributes or noun (e.g. "A woman with blue hat" vs. "A man with red hat"). Hence both images remain relatively consistent, semantically and spatially.
> - While CounterCurate is originally a multiple-choice VQA benchmark, the answer options are natural sentences, which makes it suitable for testing model's free-form QA. We reformulate the task into a captioning setup: we ask the model to generate an overview description of the image focusing on the main entity, and treat the correct choice as the ground-truth caption. We then use GPT-4o as a judge to decide whether the model’s description is closer to the ground-truth caption or to the counterfactual one.
> - We randomly sample 500 image-question pairs from the dataset and use the corresponding 500 counterfactuals as the perturbed test set. As discussed in the footnote on p. 8, in this setting we cannot expect the clean model to perform better on the counterfactual set as the counterfactuals need not be easier than the original; instead, we look at how the performance gap between original vs. counterfactual changes under contamination. The results are provided below:
>
> | **Model**   | **Epoch** | **Train (%)** | **Test (%)** | **Delta (%)** |
> |------------|-----------|----------------|------------------|---------------|
> | **LLaVA**  | –         | 72.20          | 76.87            | +4.68         |
> | **LoRA**   | 1         | 89.44          | 88.65            | -0.79         |
> |            | 2         | 89.66          | 88.44            | -1.22         |
> |            | 3         | 90.30          | 87.79            | -2.51         |
> | **LLM**    | 1         | 87.72          | 88.87            | +1.15         |
> |            | 2         | 88.79          | 86.72            | -2.07         |
> |            | 3         | 90.73          | 86.94            | -3.79         |
> | **LLM+MLP**| 1         | 87.72          | 88.01            | +0.29         |
> |            | 2         | 89.44          | 86.08            | -3.36         |
> |            | 3         | 89.87          | 85.84            | -4.03         |
> | **ALL**    | 1         | 87.72          | 88.87            | +1.15         |
> |            | 2         | 90.09          | 86.94            | -3.15         |
> |            | 3         | 90.30          | 85.87            | -4.43         |
>
> - Since this is a free-form language generation task with an unlimited output space, evaluated by a judge rather than exact matching over a fixed set of options, the signal that can be used to detect contamination can be weaker. As a result, we see some detection failures under light contamination (standard fine-tuning for 1 epoch with LLM, LLM+MLP, ALL), while these settings become detectable when contamination is stronger. Across all configurations, contaminated models still show a larger degradation between the original and counterfactual sets than the clean model, indicating that the core principle of our pipeline extends to the free-form setup.
>
>
> References:
>
> [1] CounterCurate: Enhancing Physical and Semantic Visio-Linguistic Compositional Reasoning via Counterfactual Examples, https://arxiv.org/abs/2402.13254

---

> ### Author Response · Authors · 2025-11-23
>
> **(W3) Difficulty of the perturbed problems**:
> - We first clarify that we do not require perturbed samples to be strictly easier; we only need them to be of similar (or non-harder) difficulty compared to the original question. Under this condition, we expect genuinely clean models that have learned to generalize to perform similarly on original and perturbed samples, whereas contaminated models that have merely memorized specific instances will fail on the perturbed ones (L97–103).
>     - An analogy is a very simple tweak of past exam questions in a way the question appears virtually the same but the correct answer changes. A student who truly understands the material will handle both version whereas a student who simply memorized the old answer is much more likely to fail on the tweaked one, without requiring that the tweaked question is easier than the original.
> - Our results on NaturalBench (Table 4) already illustrate that we can detect contaminated models even without any formal guarantee that the counterfactual image-question pairs are easier. In that setting, clean models remain robust under the counterfactual perturbations, while contaminated models exhibit larger drops. When the perturbed dataset is known to contain easier samples, the signal becomes even cleaner since performance gains and drops can be used directly as a detection criterion.
> - The non-harder assumption is induced from our construction. As mentioned in L323, we assume that preserving the original question and only altering the answer choice does not alter the question's difficulty. Although we agree that this assumption may not hold for certain questions, we believe it is a reasonable assumption to make given that RealWorldQA and MMStar primarily consist of simple VQA problems.
> - Lastly, the 'black-box' assumption from Requirement 1 refers to the fact that our detection method does not require any usage of prior ground-truth knowledge about the behaviors of clean models. Methods that must be calibrated using known clean models to adjust the decision threshold would violate this requirement, as depicted in Table 2 and 12. The semantic perturbation pipeline is the construction step and is not directly related to the black-box assumption described as the first Requirement.
>
> **(Q1) Guaranteeing difficulty of perturbed problems**
> - Thank you for raising this insightful question. We agree that just like how Flux+ControlNet occasionally generates easier variants by enlarging the stop sign, the opposite could also happen, leading to false positives. **Empirically, when we were manually validating the perturbed questions, we did not observe any such cases.**
> - A critical design choice that prevents generating more difficult problems is conditioning the caption generation on the original question, the original answer, and the new correct answer, and prompting the captioning model to generate captions to include all the necessary details to solve the question (L199-204, 355-365).
> - We initially tested generation by simply instructing the model to change the image so the correct answer to the question is the new answer. When evaluated on this version of the generated images, the clean model's performance was much lower compared to both the original dataset and our final perturbed dataset, with invalid perturbations appearing much more frequently, for example, due to critical component of the image being left out so the question becomes ambiguous or unsolvable.
> - Intuitively, the captioning model first reasons about which parts of the image need to change to make the new answer correct while preserving the rest of the scene, and produces a detailed caption emphasizing those changes. Flux+ControlNet then focuses on rendering exactly these components. We believe this two-stage, question- and answer-conditioned design is what mitigates the risk of perturbations unintentionally increasing difficulty.
> - More details have been updated in the main draft regarding conditioning the caption generation in Section 6.
>
> **(Q2) Practicality in lower-resource setting:**
> - As in our response to Reviewer Duhn's (W1), the cost of our perturbation pipeline is cheap with moderate hardware assumptions, and scales linearly with the number of data. More concretely, we have shown in Section 7 that GPT-4o can be replaced with a lightweight open-source model (Molmo-7B-D) and manual filtering can be replaced with o3 model. This shows that our pipeline is modular and can be adapted depending on the available resources.
> - In cases where the real-world counterfactuals already exist, we verified that our method can be applied without having to generate the perturbed datasets from strong generative models (NaturalBench, Table 4).
> - Finally, as addressed in our response to (W3), we would like to clarify once again that Requirement 1 is not about computational resources or scalability.
>
> We hope these clarify the soundness and applicability of our perturbation pipeline.

---

### Official Review · Reviewer_Duhn · 2025-10-31

**Soundness:** 3
**Presentation:** 3
**Contribution:** 3
**Rating:** 6
**Confidence:** 3

**Summary:**

This paper introduces multi-modal semantic perturbation as a detection framework for identifying data contamination in VLMs. The method detects contamination by measuring the performance degradation of a model on perturbed samples compared to the original. Experiments show the proposed detection setting is reliable.

**Strengths:**

1. Well-motivated problem definition.
2. Extensive comparisons across contamination settings (fine-tuning, LoRA, pretraining) and baselines demonstrate consistent detection performance.
3. The detection approach is straightforward by comparing performance on the original vs. perturbed input.

**Weaknesses:**

Perturbation generation requires LLM and diffusion inference per sample, implying high computational cost. This method could be a bit difficult to generalize due to scalability and efficiency constraints.

**Questions:**

See weakness. How computationally expensive is the proposed method?

---

> ### Author Response · Authors · 2025-11-23
>
> We sincerely thank Reviewer Duhn for recognizing the well motivated problem and the extensive and consistent experiments. The reviewer’s main concern is regarding the computational cost and scalability of our approach, which we address below.
>
> **(W1) Our method is linear and scalable**:
> - We first point out that our semantic perturbation pipeline is run once to construct the perturbed benchmark and then reused, and its cost scales linearly with the number of datapoints in a dataset.
> - **Caption generation**:
>     - Using GPT-4o as the captioner requires on average ~75 input tokens and ~125 output tokens per image. Generating 1,000 captions costs less than $1.50. This stage also does not require proprietary models: it can be replaced with Molmo-7B-D (Table 10), which fits on a single 24GB GPU.
> - **Flux+ControlNet generation**:
>     - Flux+ControlNet generation runs on a single 40GB GPU. With 25 sampling steps, each image takes around 15~20 seconds to generate, and this can be further reduced by decreasing the number of steps. This stage is trivially parallelizable across multiple GPUs or machines if higher throughput is needed.
> - **Automated filtering with o3**:
>     - Automated filtering requires on average ~75 input tokens and ~200 output tokens per image. Generating 1,000 filtering decisions costs less than $2.00.
> - Overall, the total cost to construct the perturbed benchmark is cheap with modest hardware requirements. Our approach scales linearly with dataset size, making our approach practical and scalable even for reasonably large benchmarks. Finally, when real-world counterfactuals are already available as part of the benchmark design (e.g., NaturalBench in Table 4), we do not need to generate perturbations at all.
>
> We hope our response addresses the reviewer's concern regarding scalability and efficiency.

---

### Official Review · Reviewer_55ak · 2025-10-31

**Soundness:** 2
**Presentation:** 2
**Contribution:** 2
**Rating:** 2
**Confidence:** 4

**Summary:**

This paper studies test-set leakage detection for VLMs. The authors deliberately “contaminate” open-source models like Qwen2-VL by fine-tuning them on benchmark data and show that prior text-based contamination detection methods fail. They then propose a method called multi-modal semantic perturbation, which uses GPT-4o to rewrite captions and Flux + ControlNet to generate visually altered images that subtly change the correct answer. If a model performs well on the original but fails on the perturbed version, it is flagged as potentially contaminated.

**Strengths:**

1. The paper targets data leakage in multimodal models, which is an important issue.

2. It provides systematic experiments across multiple contamination types and model sizes, with quantitative comparisons to several baselines.

3. The overall framework is simple and easy to understand, and the perturbed benchmark will likely be useful to the community.

**Weaknesses:**

**1. Inaccurate and heavy generation pipeline with high manual cost.**

I find the proposed “detection pipeline” is simple, unnecessarily complex, and fragile for what it aims to do. It requires GPT-4o to generate dense captions, a Flus plus ControlNet for Canny-guided editing (a very crude method for composition-preserved image generation), and finally human filtering to discard failed generations.

In addition, the method needs manual filtering to remove a large portion of the data. In particular, RealWorldQA is reduced from 765 to 440 samples, and MMStar from 1500 to 495, over 1/3 is filtered.

To me, this already contradicts their claim of practicality: a method that needs strong proprietary models and heavy human effort cannot serve as a scalable detector. The “clean vs. contaminated” separation is only visible after extensive manual curation.

**2. The detection signal is confounded with robustness, not contamination.**

The method assumes that if a model fails on perturbed images, it flags contamination. But the paper itself shows that many perturbations accidentally change task difficulty. In Figure 3, the perturbation even enlarges the visual cue (a speed limit sign), making the new image easier.

In other cases (Figure 4), the perturbed image drifts too far from the original, so even a clean model can fail while a contaminated one may still succeed. In my view, the metric mainly captures distribution shift sensitivity rather than true memorization, so the signal is not clean. Essentially, this is caused by the instability of the perturbation method being constructed.

**3. Unrealistic and self-serving contamination setup.**

The contamination experiment is extremely idealized: directly fine-tune the model on the full test set for one to three epochs. This guarantees that the model has seen every evaluation item. Real leakage in practice is much subtler: partial overlap, web-scale data, paraphrased variants, and it’s unclear whether the proposed metric would still detect that.

Therefore, while the results look strong, the paper proves only that the method works under a trivially detectable contamination pattern. It’s closer to a sanity check than to a general detection method.

**4. Limited applicability and reliance on strong visual grounding.**

The method only works for benchmarks where the image strictly determines the answer. As they note themselves, “if a question can be answered without visual input, perturbing the image is meaningless”. This excludes open-ended captioning, grounded reasoning, or OCR-heavy tasks. So the claimed “general framework for contamination detection” is actually limited to multiple-choice VQA tasks with strong visual dependency.

**Questions:**

Please see the weaknesses.

---

> ### Author Response · Authors · 2025-11-23
>
> We thank the reviewer for acknowledging the **‘systematic experiments’** and the **‘usefulness of the perturbed benchmarks.’** The reviewer’s concerns are focused on detection pipeline and experimental settings, and we address each of them below.
>
> **(W1) Accuracy and cost of the pipeline**
> - It is valid to raise concerns about the cost of the pipeline, including the manual effort in the filtering process.
>     - **Computational cost of the pipeline**: our pipeline scales linearly with the data and is cheap with moderate hardware requirements, as listed in our response to Reviewer Duhn's (W1).
>     - ControlNet is explicitly trained to preserve the original image’s composition. However, since image generation models are still imperfect, the verification stage is necessary to ensure that the perturbed questions remain solvable, preventing propagation of error into the evaluation.
>     - **High manual cost:** this is indeed an important concern, but we already showed in Section 7 that it can be avoided using the o3 model. Importantly, we verify that 86% of the images from the o3 filtered set overlap with the manually filtered set, indicating high agreement (L470-473). This confirms that our method remains effective even without any manual intervention.
>     - Moreover, with the NaturalBench experiment in Table 4, we demonstrate that when real-world counterfactual images are available, they can be used directly instead of generating perturbed benchmarks at all. In such settings, our method only requires running models on existing data, eliminating both generation and manual filtering costs.
> - We also emphasize that the filtered dataset is a representative subsample of the full generated set, as shown in Table 3. Thus, we respectfully disagree with the implication that the “clean vs. contaminated” separation is an artifact only visible after manual curation. On the contrary, including unfiltered invalid questions would confound performance differences and obscure the separation, leading to an inaccurate picture of model behavior.
> - Finally, we would like to note that practicality in Requirement 1 is not about the computational resources, but refers to the fact that our detection method does not require any usage of prior ground-truth knowledge about the behaviors of clean models, as described in our response to Reviewer 6vCV's (W3) in the last bullet point).
>
> **(W2) Confounding with robustness**:
> - First, we acknowledge that Figure 3 were samples that needed to be filtered out. After perturbation, the correct answer must change, but the image in Figure 3 was an inconsistent generation case where the correct answer did not change. We have updated the manuscript with a similar but correct example, and apologize for any confusion.
> - However, regarding the reviewer's original comment about perturbation leading to an easier variant of the original question (as was shown in the original Figure 3 as well), **we stress that this is a desired behavior, not a flaw (L210-219)**. For clean models that truly generalize, if the models could answer the original question, we should expect it to be able to answer an easier variant of the question as well. If a contaminated model solves the original problem but fails on the easier variant, it is a strong indication that the model has memorized the original question.
> - Figure 4 illustrates a rare failure case of our pipeline where the visual details change significantly without making the question easier, so that even a contaminated model can answer both variants correctly, leading to incorrect detection results. Also, we acknowledge that a clean model may fail to answer the perturbed version while answering the original question correctly, due to the (clean) models being imperfect. For example, it may be a case that the model was getting the original question right but with low confidence or purely by chance, and may fail to answer the perturbed version incorrectly. In this work, however, we assume clean models with reasoning capabilities are unlikely to suffer from this problem (L216-218).
> - Furthermore, we manually inspect the perturbed images after filtering and verify such failure cases where the visual details differ significantly are very rare. We found 8 out of 440 images (\~1.8%) from perturbed RealWorldQA and 17 out of 495 images (\~3.4%) from perturbed MMStar, which have deviated from the original question's visual details. We also note that for clean models, we always observe an increase in performance on the perturbed benchmarks (Table 2 and 12), verifying our inspection results.
> - We are testing the models under perturbations that the models are robust against. The increase in performance for clean models clearly indicate this. Also, the fact that the perturbation techniques from existing detection methods fail indicate that the both the clean and contaminated models are robust to those perturbations. Please also refer to our response to Reviewer T6b8's (W4).

---

> ### Author Response · Authors · 2025-11-23
>
> **(W3) Contamination setup**:
> - **Partial overlap**: we emphasize that our detection method still works at a sample level. Please refer to the second bullet point of our response to Reviewer T6b8’s (W2). Hence our method can provide strong signals in partial overlap cases as well.
> - **Robustness to paraphrasing:** this is an excellent suggestion and a challenging setting for contamination detection. We verify that our method still can detect models contaminated with paraphrased questions with the perturbed benchmark we generate.
>     - Concretely, we prompt GPT-4o to generate three possible paraphrases of the original question, and select the one with the **lowest 5-gram** overlap with the original question, enforcing the model to use different sentence structures and vocabularies.
> - Contaminated models, which are fine-tuned with the paraphrased version of the benchmark are evaluated on the original benchmark along with the perturbation of the original benchmark (denoted RQA and RQA_P). We first observe that models trained on the paraphrased variant show inflated performance on the original benchmark, verifying the soundness of our paraphrasing pipeline. We also consistently observe that the contaminated models perform worse on the perturbed benchmark, despite the question being phrased differently.
>
> The results for RealWorldQA are listed below:
>
> | **Model** | **Epochs** | **RQA (%)** | **RQA_P (%)** | **Delta (%)** | **Model** | **Epochs** | **RQA (%)** | **RQA_P (%)** | **Delta (%)** |
> |-----------|------------|-------------|----------------|----------------|-----------|------------|-------------|----------------|----------------|
> | **LLaVA** | – | 52.05 | 56.36 | +4.31 | **Qwen2-VL** | – | 70.45 | 71.59 | 1.14 |
> | **LoRA** | 1 | 52.03 | 40.00 | -12.03 |  | 1 | 72.50 | 71.59 | -0.91 |
> |         | 2 | 55.91 | 40.00 | -15.91 |  | 2 | 73.86 | 69.55 | -4.31 |
> |         | 3 | 59.09 | 38.86 | -20.23 |  | 3 | 75.23 | 69.32 | -5.91 |
> | **LLM** | 1 | 56.14 | 54.09 | -2.05 | **LLM** | 1 | 78.41 | 53.86 | -24.55 |
> |         | 2 | 60.91 | 49.09 | -11.82 |  | 2 | 87.05 | 56.59 | -30.46 |
> |         | 3 | 63.64 | 50.45 | -13.19 |  | 3 | 89.55 | 54.32 | -35.23 |
> | **LLM+MLP** | 1 | 56.14 | 53.41 | -2.73 | **LLM+MLP** | 1 | 80.00 | 52.50 | -27.50 |
> |            | 2 | 61.36 | 48.64 | -12.72 |  | 2 | 87.05 | 56.59 | -30.46 |
> |            | 3 | 63.64 | 49.77 | -13.87 |  | 3 | 88.64 | 55.23 | -33.41 |
> | **ALL** | 1 | 55.91 | 53.64 | -2.27 | **ALL** | 1 | 75.45 | 51.82 | -23.63 |
> |         | 2 | 62.05 | 48.86 | -13.19 |  | 2 | 86.36 | 56.36 | -30.00 |
> |         | 3 | 63.86 | 50.00 | -13.86 |  | 3 | 90.23 | 55.45 | -34.78 |
>
> The results for MMStar are listed below:
>
> | **Model** | **Epoch** | **MMStar (%)** | **MMStar_P (%)** | **Delta (%)** | **Model** | **Epoch** | **MMStar (%)** | **MMStar_P (%)** | **Delta (%)** |
> |-------------------|-----------|----------------|-------------------|----------------|------------------------|-----------|----------------|-------------------|----------------|
> | **LLaVA** | – | 37.78 | 68.29 | +31.51| **Qwen2-VL** | – | 62.02 | 78.18 | +16.16 |
> | **LoRA** | 1 | 51.52 | 43.64 | -7.88 |  | 1 | 78.87 | 67.48 | -11.39 |
> |         | 2 | 59.60 | 41.82 | -17.78 |  | 2 | 83.33 | 64.95 | -18.38 |
> |         | 3 | 62.22 | 40.81 | -21.41 |  | 3 | 85.70 | 63.74 | -21.96 |
> | **LLM** | 1 | 47.47 | 45.68 | -1.79 | **LLM** | 1 | 89.09 | 60.61 | -28.48 |
> |         | 2 | 53.13 | 50.39 | -2.74 |  | 2 | 96.36 | 55.76 | -40.60 |
> |         | 3 | 61.82 | 53.37 | -8.45 |  | 3 | 97.98 | 54.34 | -43.64 |
> | **LLM+MLP** | 1 | 47.27 | 46.06 | -1.13 | **LLM+MLP** | 1 | 87.27 | 61.62 | -25.65 |
> |            | 2 | 57.60 | 53.54 | -3.06 |  | 2 | 97.17 | 55.96 | -41.21 |
> |            | 3 | 62.42 | 57.98 | -4.44 |  | 3 | 97.78 | 56.36 | -41.42 |
> | **ALL** | 1 | 46.87 | 45.25 | -1.62 | **ALL** | 1 | 80.20 | 63.64 | -16.56 |
> |         | 2 | 53.33 | 49.52 | -3.81 |  | 2 | 94.95 | 54.95 | -40.00 |
> |         | 3 | 62.42 | 57.58 | -4.84 |  | 3 | 96.77 | 55.15 | -41.62 |
>
> - The results on the tables demonstrate that our pipeline reliably detect all models that have been contaminated with paraphrases of the questions, proving robust against such possible attacks. This will be updated on the manuscript.
> - **Web-scale data**: Simulating contamination during pretraining stage with web-scale data is often impossible not only due to the required amount of compute but also because the pretraining corpora is often proprietary. However, for LLaVA models, the pretraining corpus is publicly available and we have already simulated a scenario where the test set has leaked during the pretraining stage, and verified that our method can detect contamination (Table 6).

---

> ### Author Response · Authors · 2025-11-23
>
> - **More realistic contamination scenario:** To simulate a more realistic contamination scenarios, we fine-tune models with a mixture of 6 popular multi-modal benchmarks: MathVista[1], MMBench[2], MMMU[3], CV-Bench[4], MMStar and RealWorldQA, resulting in 11,280 image-question pairs. We test whether our approach can still detect contamination on RealWorldQA and MMStar, which now consist only of \~6.7% and \~13.3% of the fine-tuning dataset, respectively. The results are listed below:
>
> | **Model** | **Epoch** | **RQA (%)** | **RQA_P (%)** | **Delta (%)** | **Model** | **Epoch** | **RQA (%)** | **RQA_P (%)** | **Delta (%)** |
> |-----------|-----------|-------------|---------------|---------------|-----------|-----------|-------------|---------------|---------------|
> | **LLaVA** | – | 52.05 | 56.36 | +4.31 | **Qwen2-VL** | – | 70.45 | 71.36 | +0.91 |
> | **LoRA** | 1 | 45.91 | 41.59 | -4.32 |  | 1 | 72.95 | 70.00 | -2.95 |
> |  | 2 | 62.73 | 40.23 | -22.50 |  | 2 | 75.91 | 68.18 | -7.73 |
> |  | 3 | 75.45 | 42.05 | -33.40 |  | 3 | 76.14 | 68.31 | -7.83 |
> | **LLM** | 1 | 59.55 | 52.27 | -7.28 | **LLM** | 1 | 77.50 | 50.00 | -27.50 |
> |  | 2 | 65.68 | 50.23 | -15.45 |  | 2 | 85.00 | 45.68 | -39.32 |
> |  | 3 | 67.05 | 46.14 | -20.91 |  | 3 | 88.18 | 47.95 | -40.23 |
> | **LLM+MLP** | 1 | 57.95 | 52.05 | -5.90 | **LLM+MLP** | 1 | 75.45 | 53.41 | -22.04 |
> |  | 2 | 61.14 | 50.91 | -10.23 |  | 2 | 85.45 | 48.18 | -37.27 |
> |  | 3 | 64.32 | 48.86 | -15.46 |  | 3 | 86.36 | 47.50 | -38.86 |
> | **ALL** | 1 | 60.00 | 51.82 | -8.18 | **ALL** | 1 | 78.64 | 50.00 | -28.64 |
> |  | 2 | 65.91 | 48.86 | -17.05 |  | 2 | 87.73 | 46.59 | -41.14 |
> |  | 3 | 67.95 | 46.82 | -21.13 |  | 3 | 89.32 | 46.59 | -42.73 |
>
> | **Model** | **Epoch** | **MMStar (%)** | **MMStar_P (%)** | **Delta (%)** | **Model** | **Epoch** | **MMStar (%)** | **MMStar_P (%)** | **Delta (%)** |
> |-----------|-----------|-------------|---------------|---------------|-----------|-----------|-------------|---------------|---------------|
> | **LLaVA** | – | 37.78 | 68.29 | +31.51 | **Qwen2-VL** | – | 62.02 | 78.18 | +16.16 |
> | **LoRA** | 1 | 45.86 | 26.67 | -19.19 |  | 1 | 78.38 | 71.31 | -7.07 |
> |  | 2 | 83.43 | 54.14 | -29.29 |  | 2 | 94.14 | 65.25 | -28.89 |
> |  | 3 | 85.25 | 52.93 | -32.32 |  | 3 | 95.96 | 63.64 | -32.32 |
> | **LLM** | 1 | 42.73 | 38.28 | -4.45 | **LLM** | 1 | 89.90 | 60.40 | -29.50 |
> |  | 2 | 62.02 | 35.15 | -26.87 |  | 2 | 97.98 | 54.95 | -43.03 |
> |  | 3 | 73.54 | 41.82 | -31.72 |  | 3 | 98.99 | 55.96 | -43.03 |
> | **LLM+MLP** | 1 | 34.65 | 33.03 | -1.62 | **LLM+MLP** | 1 | 90.10 | 61.41 | -28.69 |
> |  | 2 | 62.42 | 38.59 | -23.83 |  | 2 | 97.17 | 55.56 | -41.61 |
> |  | 3 | 82.22 | 48.48 | -33.74 |  | 3 | 98.79 | 55.96 | -42.83 |
> | **ALL** | 1 | 32.22 | 30.41 | -1.81 | **ALL** | 1 | 85.86 | 62.83 | -23.03 |
> |  | 2 | 77.78 | 45.05 | -32.73 |  | 2 | 89.70 | 57.37 | -32.33 |
> |  | 3 | 81.82 | 45.45 | -36.37 |  | 3 | 93.74 | 55.76 | -37.98 |
>
> - The tables above clearly demonstrate that our method can reliably detect contaminated models even when the contamination signal is weaker, resembling a more realistic fine-tuning contamination scenario.
> - **Focus on fine-tuning**: We note in L174 that it is computationally infeasible to conduct exhaustive experiments without assuming contamination during fine-tuning stage. Therefore, we mainly focus on this controlled setup, but ablate more realistic scenarios including (i) leakage during pre-training stage when possible (Table 6), (ii) contamination with paraphrased data and (iii) contamination with a mixture of other benchmarks. The consistent detection results across these experiments verify that our method is not tied to a single "self-serving" contamination design, but extends to more realistic and subtle forms of leakage that are not "trivially detectable contamination patterns."
>
> References:
> [1] MathVista: Evaluating Mathematical Reasoning of Foundation Models in Visual Contexts, https://arxiv.org/abs/2310.02255
> [2] MMBench: Is Your Multi-modal Model an All-around Player?, https://arxiv.org/abs/2307.06281
> [3] MMMU: A Massive Multi-discipline Multimodal Understanding and Reasoning Benchmark for Expert AGI, https://arxiv.org/abs/2311.16502
> [4] Cambrian-1: A Fully Open, Vision-Centric Exploration of Multimodal LLMs, https://arxiv.org/abs/2406.16860

---

> ### Author Response · Authors · 2025-11-23
>
> **(W4) Reliance on visual grounding**:
> - Our focus in this work is on truly multi-modal tasks that require both the visual and textual input to answer the question, and we establish a principled framework that can be used to detect VLMs that have been contaminated on such tasks.
> - Our understanding is that the reviewer is referring to open-ended (image) captioning, (visually) grounded reasoning, or OCR-heavy (image understanding) tasks, which are all tasks that cannot be answered without visual input. Our pipeline applies to these benchmarks as long as (i) the answer is tied to concrete visual content and (ii) we can construct semantically perturbed images that flip the correct answer while preserving overall composition. To verify that our task can be applied beyond the multiple-choice settings, we have run our pipeline under free-form QA and verified that our pipeline still detects contamination. Please refer to the table in our response to Reviewer 6vCV’s (W2).
> - Finally, we respectfully contend that the reviewer has misquoted our manuscript, and note that our manuscript does not claim a "general framework for contamination detection." We state that we propose an “effective detection framework based on multi-modal semantic perturbation” (L110) to identify contaminated models by “testing for generalization failures in the visual domain” (L112). Our claims are explicitly scoped to visually grounded evaluation and we have made no claims that our framework can generalize to any or all contamination detection tasks.
>
> We hope that these points clarify the our problem and experiment setting, and demonstrate its practicality and validity in various contamination scenarios.

---

### Official Review · Reviewer_T6b8 · 2025-11-03

**Soundness:** 2
**Presentation:** 2
**Contribution:** 2
**Rating:** 4
**Confidence:** 4

**Summary:**

The authors propose a novel detection framework based on multi-modal semantic perturbations, which involves generating new test samples by subtly altering an image’s content (while preserving overall semantics) so that a model which merely memorized the original image-text pair will fail on the perturbed input. Experiments show that existing detection methods often fail or give inconsistent results on VLMs, whereas the proposed perturbation-based approach consistently flags contaminated models across diverse fine-tuning settings and degrees of contamination, satisfying key requirements of reliability, practicality, and consistency.

**Strengths:**

Proposes an original and practical contamination detection method tailored for VLMs using multi-modal semantic perturbations.

Demonstrates strong technical quality through extensive and controlled experiments across diverse settings.

Clear presentation with significant implications for reliable and fair evaluation of vision-language models.

**Weaknesses:**

While the proposed method is practically useful, it primarily consists of integrating existing tools—LLMs for captioning and diffusion models for image editing—into a contamination detection pipeline. As such, the technical novelty is relatively limited. The idea of testing generalization via perturbed inputs is well-established, and the paper applies this concept to the multi-modal setting without introducing fundamentally new algorithms or theoretical insights.

The description of the core methodology in Section 4 is relatively high-level and omits several important implementation and design details. For example, while the use of GPT-4o and ControlNet is outlined, it is unclear how semantic alignment between the new answer and generated image is ensured, or how failure cases are handled systematically beyond manual filtering.

The core contamination detection criterion—declaring a model contaminated if it answers perturbed samples incorrectly while answering original ones correctly—is intuitive but lacks technical precision. The paper does not clarify whether this evaluation is done at the sample level, across aggregate performance metrics, or via some probabilistic threshold.

Since the proposed method relies on detecting failures under semantic perturbations, it would be informative to compare against standard OOD generalization or robustness baselines. This would help disentangle contamination from general lack of robustness.

**Questions:**

same as weakness

---

> ### Author Response · Authors · 2025-11-23
>
> We sincerely thank Reviewer T6b8 for recognizing the **significant implications** of our work for reliable evaluation of VLMs, and highlighting the **originality, practicality and extensive experiments.** We address each of the reviewer’s concerns below.
>
> **(W1) Technical Novelty:**
> - **The primary contribution of this work is a perturbation methodology specifically designed for VLMs** that operates under extensive and systematic contamination scenarios. While testing generalization via perturbed inputs is a common theme in prior contamination detection literature, we show that existing input-perturbation techniques break down in the multi-modal setting and we instantiate a concrete, effective perturbation strategy tailored to VLMs.
> - **Extending uni-modal techniques to the multi-modal space is inherently non-trivial.** For instance, simple masking strategies that work for LLMs do not transfer, as the continuous and highly entangled nature of image space makes it difficult to selectively remove spurious cues without altering task semantics or difficulty. Our method explicitly addresses this challenge by constructing perturbations that change the underlying semantics in a controlled manner while preserving overall composition.
> - Among existing VLM contamination works (L513–519), Lu et al. (2024a) mitigate spurious cues by shuffling color channels, and Song et al. (2025) introduce image masking and option shuffling to test robustness. However, such perturbations cannot reliably identify contaminated VLMs, as performance drops may arise from artifacts or sampling bias rather than genuine memorization effects. In contrast, our framework detects contamination consistently across fine-tuning regimes (Table 2, 12), requires no prior knowledge of which models are contaminated, and yields performance changes that closely track the true contamination level (Table 2, 12)
> - Our pipeline can be viewed as an instance of generalization-based detection methods (L264–269). To the best of our knowledge, however, we are the first to systematically modify the question–image pair so that the correct answer itself changes, rather than merely perturbing surface cues. To enable this, we develop a principled pipeline that combines existing generative tools with a carefully designed perturbation and evaluation scheme into a coherent, effective framework. This design also passes extensive ablations, including real-world counterfactual perturbations (Sec. 7, L381–386), further supporting that our approach is not a heuristic assembly of components but a principled and robust method for multi-modal contamination detection.
>
> **(W2) Methodological Detail:**
> - To clarify, Appendix A.2 and B already provide full details, including the exact system prompt that we use for caption generation. If the reviewer could be more specific about which parts of the pipeline description were unclear, we would be happy to address them in more detail, and update the manuscript accordingly.
> - We stress that both the pipeline design and the model choices are explicitly aimed at enforcing semantic alignment as much as possible. To ensure that the newly generated images’ components are spatially aligned with the original ones, we use ControlNet, which is specifically designed to edit images while keeping the original composition intact. Specifically, we use Flux+ControlNet, since Flux is known to process long text input and generate images that are semantically aligned with the prompt (for examples, see https://www.giz.ai/flux-1-prompt-guide/). To ensure that the overall semantics within the image remain consistent with the new answer, we first generate a dense caption of the image and feed this caption as the input prompt to Flux+ControlNet, so that the model has access to a detailed, scene-level description rather than a short, underspecified instruction.
> - Because current image generation models are still imperfect, we applied manual filtering to demonstrate the upper-bound performance of our approach. However, as shown in Table 7, this manual step can be replaced with automated filtering using o3. Importantly, we verify that 86% of the images from the o3-filtered set overlap with the manually filtered set, indicating high agreement (L470–473). The results obtained with o3 confirm that our method remains effective even without any manual intervention, and that failure cases can be handled systematically rather than ad hoc, demonstrating the robustness and practical applicability of our pipeline.

---

> ### Author Response · Authors · 2025-11-23
>
> **(W3) Contamination Criterion:**
> - Thank you for raising this point. As briefly mentioned in L212-216, our detection method will work on a sample level, but the main results show dataset-level detection results using aggregate performance metrics. We have clarified in the revision in Section 4 and 5 that our method can detect contamination both at the sample level and at the dataset level using aggregate performance metrics.
> - **For sample-level detection:** we consider pairs where the model answers the original question correctly but fails on a perturbed version that is comparable or easier in difficulty. In this case, the failure on the perturbed sample - despite preserved or reduced difficulty - serves as evidence that the model may have memorized the original instance rather than learning to generalize. In our experiments, we observe performance drops as large as -45% under such perturbations, indicating that per-sample inspection can provide a strong contamination signal.
> - **For dataset-level detection:** as shown in the main paper, we compare how the model's accuracy on the original vs perturbed dataset changes to determine contamination. We treat this discrepancy as signal rather than probabilistic approaches.
>
> **(W4) Contamination vs. Robustness/OOD Generalization:**
> - This is a good suggestion. In fact, we already showed this in the main paper.  Specifically, we always show the baseline performance of the clean model (which has not been contaminated) on both the original and perturbed datasets. We can clearly see from Tables 2, 5, 8, and 12 that the clean models are robust to our semantic perturbations since the performances remain similar or improve on the perturbed datasets. In addition, we further see that the clean models are robust to natural counterfactuals, as shown in Table 4 with NaturalBench. These results verify that **we are testing under a perturbation that the original models are robust against.**
> - The contaminated models we study are finetuned from the clean models. Thus, the performance gap between clean and contaminated models reflects contamination.
> - More importantly, the failure of existing detection methods indicates that **both clean and contaminated models exhibit robustness to other perturbations** suggested in the prior approaches.
>
> We hope these clarifications address the raised concerns and make the contributions of our work more precise.

---

### Author Response · Authors · 2025-11-24

We sincerely thank all reviewers for their valuable time and feedback on our manuscript.

We have carefully considered and responded to all comments and updated our manuscript accordingly to incorporate additional clarifications and new experiment results to address the main concerns raised during the rebuttal. We have highlighted all the revisions in **red** to make the changes clear.

Below, we summarize the new additions on the updated manuscript.

1. Section 4:
- Provided extra clarification that our detection targets detecting dataset-level contamination, while it is possible to apply it to detect sample-level contamination.
2. Section 6:
- We have made our assumption regarding the clean model's behavior more explicit and updated the images in Figure 3. While the original images in Figure 3 did demonstrate a case where the perturbation yielded an easier variant, the image was sampled from the images that should be filtered, because the correct answer did not change accordingly. The images are now replaced with a similar but correct sample.
- Added more analysis to the caption generation stage.
- Added results on the frequency of failure cases in our perturbed benchmark.
3. Section 7:
- Added new results on contamination with paraphrased data.
- Added new results on contamination with a mix of 6 benchmarks.
4. Appendix:
- Added the cost and scalability analysis of our pipeline
- Added full experiment results that were omitted due to space, with 4 training strategies (LoRA, LLM, LLM+MLP, ALL) for all experiments that are conducted.

---

### Author Response · Authors · 2025-12-03
**Summary of the rebuttal**

We now summarize the individual reviewer's comments and our responses below.

- **Reviewer T6b8 (rating: 4)**
   - Limited Novelty:
       - Addressed in the response and summary above.
   - Methodological Detail:
       - Clarified the design choices of the pipeline; filtering with o3 model bypasses the manual effort.
   - Contamination Criterion:
       - Revised the paper to clarify that our detection is a dataset-level detection, but could extend to sample-level detection (L210-218).
   - Robustness:
       - Addressed in Common concerns 1.
- **Reviewer 55ak (rating: 2)**
   - Scalability and cost:
       - Addressed in Common concerns 2.
   - Robustness:
       - Addressed in Common concerns 1.
   - Contamination setup:
       - Ablation experiments demonstrate that our detection pipeline works in various more realistic scenarios, including (i) pretraining leakage (ii) mixture of 6 benchmarks and (iii) contamination with paraphrased data.
       - It is practically infeasible to perform extensive experiments at this scale without assuming contamination in fine-tuning stage.
       - We also emphasize our work assumes moderate and realistic contamination scenarios (L180-182, 507-509), and is not designed for “self-serving” as the reviewer states.
   - Limited applicability and reliance on visual grounding:
       - The reviewer **misquoted** our manuscript, as we never claim our framework to be a “general framework for contamination detection.” As explained in L110-112, 375-377, we make the scope of our work clear. We emphasize that this assumption is also critical to investigate contamination in truly multi-modal tasks that require both the visual and textual input.
- **Reviewer Duhn (rating: 6)**
   - Scalability and cost:
       - Addressed in Common concerns 2.
- **Reviewer 6vCV (rating: 6)**
   - Capability of generative models:
       - We have shown that GPT-4o can be replaced with Molmo-7B-D, and believe that it is reasonable to assume these models will remain at least as capable, if not better.
   - Free-form QA:
       - Addressed in Common concerns 3.
   - Difficulty of perturbed problems:
       - Clarified that we do not require the variants to be easier.
       - False positives (increased problem difficulty): we manually verified and did not observe any such cases.
       - Revised the draft to describe the key design choice in yielding a generalized benchmark (L358-365).
   - Practicality:
       - Clarified the misunderstanding that the black-box definition in Requirement 1 is not about resources or scalability.
       - We also show that our pipeline is cheap with moderate hardware requirements. Furthermore, GPT-4o can be replaced with Molmo-7B-D in Table 10, 22, 23 depending on the available resources.


Taken together, we believe this work makes a substantial contribution to contamination detection in VLMs and to the broader ML community. We once again thank the reviewers and the AC for their time and thoughtful considerations.

---

### Author Response · Authors · 2025-12-03
**Summary of the rebuttal**

For the Area Chair’s convenience, we briefly summarize the initial round of discussion. We sincerely appreciate the reviewers’ comments, and we believe we have fully addressed all concerns raised. The revised draft, which incorporates clarifications and new experiments, is significantly stronger and makes our contribution much clearer. Many of the original concerns, including W2-4 form T6b8 and W1-3 from 55ak stemmed from misunderstandings of our problem setting and methodology, which we have now explicitly clarified in the text. Under a normal rebuttal timeline, we are confident that these clarifications would have resolved the outstanding issues.

### **Summary of original manuscript**

Our paper tackles an important but underexplored problem of contamination detection in VLMs. We first verify that all existing approaches perform poorly to detect the contaminated models we test (experimental evaluations are provided in Sec.5). We hypothesize that this is mainly due to:
1. Moderate contamination assumption in this work (i.e. models are trained for at most 3 epochs) (L180-182, 508)
2. Extra vision modality which exhibit different characteristics from the discrete text domain (L515-519)

**both of which make the problem non-trivial.**

We then come up with a novel idea to create a variant of the problem while maintaining the image semantics, and devise a simple pipeline to enable this multi-modal perturbation. When tested on multi-modal benchmarks, we observe that our approach satisfies Practicality, Reliability and Consistency requirements (L147-153) in all of our experiments.

We also conduct an extensive analysis to understand why our pipeline works (Section 6), and perform extensive ablation studies (Section 7) and demonstrate that
1. Manual filtering can be automated (Table 9, 20, 21)
2. GPT-4o can be replaced with a light-weight open-source model during captioning stage (Table 10, 22, 23)
3. Perturbations need not be synthetic (Table 6, 19)
4. Detects contamination with varying model size during pre-training, mixture of benchmarks, and critically, with **paraphrased questions** (Table 8, 28-31, 24-27)
5. Can be extended beyond multiple choice VQA setup, given a robust evaluation scheme (Table 32)

We now briefly summarize the reviewer comments and our response below.

### **Common concerns amongst reviewers:**
   1. Robustness (Reviewer T6b8 and 55ak):
       - We report clean model performance and observe increased performance on the perturbed benchmark. This indicates that **we are testing with perturbations that models are robust against**. Contaminated models are trained from clean models which are robust to the perturbations we are testing.
       - The failure of existing detection approaches further demonstrate that **both the clean and contaminated** models are robust to naive perturbations.
   2. Scalability of the pipeline (Reviewer 55ak, Duhn and 6vCV):
       - Generating 1,000 images with GPT-4o and verifying with o3 model costs less than $3.50 total.
       - Our pipeline has moderate hardware requirements. The Flux+ControlNet generation can be sped up by tuning the sampling steps parameter from 25 to a much lower value.
   3. Extension beyond multiple-choice questions (Reviewer 55ak and 6vCV):
       - We provide results with CounterCurate on free-form VQA with evaluation from LLM-as-a-judge.
       - The contamination signal is weaker due to the nature of language generation task and imperfect evaluation from using an LLM. However, our pipeline still detects 9 out of 12 contaminated models, and the failure cases being the models trained for 1 epoch only, indicating that the **core principle extends to the free-form setup.**

---

### Meta-Review · Area_Chair_L1rB · 2026-01-07

**Summary:**

The reviewers raised several substantive concerns that influenced the decision discussion.
A central issue is whether the proposed multi-modal semantic perturbation truly isolates contamination effects, or whether the observed signal is partially or largely explained by robustness or distribution-shift sensitivity rather than memorization (raised most strongly by Reviewer 55ak, and also noted by Reviewer T6b8). This challenges the core assumption that failure under perturbation is a reliable proxy for test-set leakage.

Another major concern is the practicality and scalability of the detection pipeline. Reviewer 55ak questioned the reliance on strong proprietary models (e.g., GPT-4o, Flux + ControlNet) and the substantial manual or semi-manual filtering, arguing that this contradicts the paper’s claim of practicality as a general detection method. Reviewer 6vCV echoed this concern more moderately, framing it as dependence on future availability and reliability of strong generative tools.

Reviewers also questioned the realism of the contamination setup. Reviewer 55ak argued that fine-tuning directly on full test sets represents an idealized and self-serving leakage scenario, and that it is unclear whether the method would remain effective under subtler, web-scale or partial-overlap contamination. While Reviewer 6vCV was more positive, they still highlighted that the method relies on assumptions about perturbation difficulty that are not formally guaranteed.

Finally, there is concern about the scope of applicability. Reviewer 55ak emphasized that the method is largely limited to visually grounded, multiple-choice VQA tasks, and does not clearly extend to open-ended, OCR-heavy, or less visually determined benchmarks. Reviewer T6b8 similarly noted the lack of precision in defining the detection criterion and suggested stronger connections to robustness or OOD baselines.

Together, these concerns suggest that while the paper addresses an important and underexplored problem and provides extensive empirical evidence, questions remain about whether the proposed signal cleanly corresponds to contamination, how broadly applicable the method is, and how practical it is beyond controlled experimental settings.

**Reviewer Concerns:**

The rebuttal addresses several substantive concerns raised by the reviewers. In particular, concerns about the realism of the contamination setup (Reviewer 55ak) are partially mitigated by additional experiments on natural counterfactual benchmarks and by showing that the detection signal degrades gradually with weaker contamination. The authors also provide new evidence that the approach can extend beyond strictly multiple-choice VQA settings, responding to the scope limitations noted by Reviewer 6vCV. Concerns about heavy manual filtering and pipeline fragility (Reviewer 55ak) are alleviated to some extent by demonstrating that automated filtering produces results largely consistent with manual curation.

However, some concerns remain only partially resolved. Multiple reviewers (55ak, T6b8) questioned whether the observed detection signal cleanly reflects contamination rather than general robustness or sensitivity to semantic distribution shifts. While the rebuttal argues empirically that contaminated models exhibit larger performance gaps, the distinction between memorization and robustness is not fully disentangled. Additionally, the assumption that perturbed samples are of comparable or non-harder difficulty, highlighted by Reviewer 6vCV, is supported mainly through intuition and empirical behavior of clean models rather than by a principled guarantee. Finally, despite clarifications about the meaning of “practicality,” the reliance on strong generative models and carefully constructed perturbations continues to limit the method’s applicability as a broadly deployable contamination detector, as originally noted by Reviewer 55ak.

**Reviewer Scores:**

Reviewer 6vCV would likely remain slightly positive, as most of their concerns are addressed by additional experiments and clarifications, though some assumptions remain implicit.

Reviewer T6b8 would likely stay near the acceptance threshold; the rebuttal improves clarity and motivation but does not fully resolve concerns about formalization and novelty. Reviewer Duhn would likely make only a minor upward adjustment, as questions around practicality and robustness are partially addressed but not fully eliminated.

Reviewer 55ak would likely not change their score substantially, since their core objections regarding robustness confounding and the realism of the contamination setup remain only partially mitigated.

Overall, the rebuttal may reinforce borderline-positive reviews but is unlikely to significantly shift strongly negative ones.

---

### Decision · Program_Chairs · 2026-01-26

Accept (Poster)